# INVESTALIGN: ALIGN LLMS WITH INVESTOR DECISION-MAKING UNDER HERD BEHAVIOR

## ABSTRACT

Studying investor decision-making processes under herd behavior is of great significance in microeconomics and behavioral finance. Large Language Models (LLMs) can be leveraged to assist in solving specific classes of complex investment problems. However, the investment decisions generated by existing LLMs often deviate from real-user data. One method to align LLMs with investor decision-making processes is Supervised Fine-Tuning (SFT), which requires a substantial amount of real-user data that is costly to collect and raises concerns about privacy and security. To overcome data scarcity, in this work, we propose **InvestAlign**, a low-cost and high-quality method that constructs large-scale SFT training datasets based on the theoretical solution to a specific simpler optimal investment problem, rather than the original complex one. We theoretically demonstrate that fine-tuning LLMs with these datasets leads to faster parameter convergence compared to using real-user data. By fine-tuning LLMs, we obtain **InvestAgent**s, which align more closely with real-user data than pre-SFT LLMs in both the simple and original complex problems examined in our study. This highlights **InvestAlign** as a promising approach with the potential to address complex optimal investment problems and align LLMs with investor decision-making processes in economics and finance.

## 1 INTRODUCTION

In financial markets, investors typically make decisions based on their risk preferences to achieve higher returns, lower volatility, and maximize their utility. (Merton (1969)). Investment decisions are crucial as they not only impact individual financial outcomes but also shape market dynamics and overall economic stability, making them a key driver of both personal wealth and broader market efficiency (Ahmad & Wu (2022)). During this process, investment assistants such as financial analysts and fund managers, play a significant role by sharing their own investment decisions through platforms (Brown et al. (2008)). These investment assistants often have rich investment experience and extensive influence, leading investors to mimic their behaviors. This is commonly referred to as herd behavior in microeconomics and behavioral finance (Bikhchandani & Sharma (2000)). The prior works in (Wang & Zhao (2024a;b)) have investigated the optimal investment problem considering herd behaviors between one investment assistant and one investor, and theoretically analyzed the impact of herd behavior on investment decisions. However, there are more complex problems where the above models fall short or only provide qualitative insights, failing to offer optimal investment advice (Zhou & Liu (2022)), which prompts us to explore alternative approaches.

Large Language Models (LLMs) have been widely adopted in various domains as generative agents to assist with specific tasks (Kovač et al. (2023); Bran et al. (2023)). There is an emerging trend that LLM agents are equipped with human-like intelligence to simulate human decision-making processes (Gao et al. (2023)). In economics and finance, substantial works have been done on aligning LLMs with human values and decisions, particularly in models for market behavior prediction and the analysis of complex economic data for policy-making (Zhao et al. (2023); Lee et al. (2024)). These efforts often focus on macroeconomic issues, such as information dissemination and collective decision-making in global markets (Li et al. (2024b)). To our best knowledge, little attention has been paid to LLMs' performance in microeconomics and behavioral finance, especially concerning investor decision-making under herd behavior, and current LLMs are shown to not fully align with investors' behavior in micro-level financial decision-making, as demonstrated in Section 3.

Figure 1: Overview of **InvestAlign**.

Achieving the alignment of LLMs to investors' decision-making processes often relies on large-scale real-user data in Supervised Fine-Tuning (SFT) (Zhang et al. (2023)). Fine-tuned with specific training datasets, LLMs can better reflect investor behavior in complex problems. However, it faces the following obstacles. Collecting real-user data can be costly due to the wide variation in investors' attributes, such as risk preference and herd behavior degree (Abbot (2017)). Additionally, many investors are reluctant to share their investment decisions due to privacy and security concerns.

To address the data scarcity problem, note that for some simple problems such as the one in Wang & Zhao (2024a), we have already found its theoretical solution, using which we can generate a large amount of data. Therefore, one possible solution is, given a complex problem, we first identify a similar and simpler problem with a theoretical solution, construct the SFT training dataset using this theoretical solution, and then fine-tune LLMs to solve the original complex problem. There are several issues to be addressed when following this approach:

**Question A**: Given the complex problem, how to identify a similar and simpler problem? What are the requirements and constraints?

**Question B**: Do the theoretical solution of the simpler problem align with real users' investment decisions, and can they be used to construct a training dataset that mirrors investor decision-making processes?

**Question C**: How can we generate the training dataset based on the theoretical solution of the simpler problem? How does it perform in aligning with investors' decision-making processes compared with real-user data?

**Question D**: How to adapt the fine-tuned LLMs to solve the original complex problem, and what is its performance?

To verify the feasibility of the proposed approach and to address the above four issues, in this work, we consider the following simple scenario of optimal investment as an example. Assume that there are two agents, one is an investment assistant and the other is an investor whose investment decisions are unilaterally influenced by the assistant under the herd behavior. For the original complex problem to solve, we consider the relative herd behavior in Wang & Zhao (2024b) where the investor adjusts his/her investments in response to changes made by the investment assistant within the same time interval and imitates the changing rate of the investment assistant's decision. Note that for this problem, even though we find its theoretical solution, its computation complexity is very high. To answer **Question A**, we use the absolute herd behavior in Wang & Zhao (2024a) as the simpler problem where the investor replicates the entire portfolio of the investment assistant, and its established theoretical solution can be derived more easily. Note that the two problems are similar in their mathematical forms, while they differ in their methods of measuring herd behavior. Then, to answer **Question B**, we collect real-user data on the simpler problem using interviews and questionnaires, and apply statistical methods to validate the consistency between real-user data and the theoretical solution. Next, to answer **Question C**, we construct SFT training datasets based on the theoretical solution, and theoretically prove that fine-tuning LLMs on the above training datasets leads to faster parameter convergence than using real-user data. Then, to answer **Question D**, given the training dataset, we fine-tune the LLMs and develop the **InvestAgent**s, which can make decisions similar to the theoretical solution, thus aligning with real-user data. Finally, we conduct another real-user test to verify the performance of **InvestAgent**s on solving the original complex problem, and experimental results show that **InvestAgent**s exhibit better alignment performance than pre-SFT LLMs.

In conclusion, our contributions include: (1) we explore and utilize LLMs in finance and economics, particularly in the domain of optimal investment under herd behavior; (2) we propose the LLM alignment techniques, which construct a large amount of high-quality dataset effectively using the theoretical solution of the corresponding mathematical model, and then apply SFT to fine-tune LLMs.

## 2  RELATED WORKS

**LLMs in Finance and Optimal Investment.**  For finance-related tasks, several specialized LLMs have been developed, e.g., BloombergGPT (Wu et al. (2023)), FinGPT (Yang et al. (2023a)), and XuanYuan 2.0/3.0 (Zhang & Yang (2023)). The success of these models depends on large amounts of training data, and the challenge is how to effectively collect and generate high-quality data, which is a key goal of our proposed method. Focusing on the optimal investment problem, prior studies have explored the use of LLMs in different scenarios such as investment idea generation and quantitative investment (Li et al. (2023); Wang et al. (2023a)). However, within agent-based modeling, there are only a few works that use LLMs as generative investors to simulate or complement human investor behavior, e.g., InvestLM in Yang et al. (2023b) and EconAgent in Li et al. (2024b). Similar agent-based ideas using LLMs have been widely used in many areas such as problems in the economic system (Horton (2023); Chen et al. (2023); Geerling et al. (2023)), social science (Ghaffarzadegan et al. (2023); Liu et al. (2024); Wang et al. (2024b)), and natural science (Bran et al. (2023); Boiko et al. (2023)). While several studies in other domains have explored the LLM agents' irrational behaviors to mirror human cognitive biases (Liu et al. (2024); Wang et al. (2024a); Xiao et al. (2024)), existing agent-based LLM models for investment have not yet accounted for the herd behavior (Bikhchandani & Sharma (2000)), which is significant in microeconomics and behavioral finance. Understanding its influence on the optimal investment problem while incorporating LLMs is crucial for analyzing investor behavior (Ahmad & Wu (2022)).

**LLM Alignment.**  LLM alignment with human values has emerged as a critical area of research (Wang et al. (2024d)). AI alignment aims to make AI systems behave in line with human intentions and values (Ji et al. (2023)). Although LLMs excel in various tasks, issues like untruthful answers (Bang et al. (2023)), sycophancy (Perez et al. (2022)) and deception (Steinhardt (2023)), along with the rise of LLM-based agents (Xi et al. (2023); Wang et al. (2024c)), raise concerns about controllability and risks in advanced AI systems. To achieve forward alignment, which ensures that trained systems meet alignment requirements, numerous methods for policy learning and scalable oversight are proposed (Ji et al. (2023); Wang et al. (2024d)). For LLMs, a typical approach is reinforcement learning from human feedback (RLHF) (Christiano et al. (2017)), with extensions like reinforcement learning from AI feedback (RLAIF) (Bai et al. (2022)) and reinforcement learning from human and AI feedback (RLHAIF) (Bowman et al. (2022)). Their pipeline includes supervised fine-tuning (SFT) (Ouyang et al. (2022); Rafailov et al. (2024)). In economics and finance, only a limited number of studies involve LLMs (Li et al. (2024b); Horton (2023)), focusing on macro-level alignment while ignoring specific and microcosmic behaviors of human decision-making.

**SFT Methods in Optimal Investment.**  Supervised fine-tuning (SFT), also known as instruction tuning, is a widely adopted technique in the field of LLMs for improving model performance on specific tasks by refining pre-trained models with a dataset tailored to the target task (Zhang et al. (2023)). Many tricks and methods of SFT have been proposed to achieve better preference alignment of LLM to humans, e.g., Ding et al. (2023); Wang et al. (2023b); Xie et al. (2024); Li et al. (2024a). In the domain of finance, SFT has been applied to various investment-related tasks such as stock prediction, financial reports summarization, sentiment analysis, portfolio optimization, etc. (Zhao et al. (2024); Guo & Hauptmann (2024); An et al. (2024)). These advancements highlight the power of SFT in tailoring LLMs to meet the specific needs of investment strategies, enabling models to simulate or complement human-like behaviors. However, collecting large, high-quality datasets for fine-tuning in optimal investment remains a challenging problem (Abbot (2017)).

## 3  PROBLEM SIMPLIFICATION AND REAL-USER DATA VERIFICATION

To verify the feasibility of the proposed method **InvestAlign**, we consider the simple scenario of optimal investment where there are two agents including the investment assistant and the investor. The original complex problem is the optimal investment under the relative herd behavior, defined as *P1* in the following (Wang & Zhao (2024b)). For the simple problem with the theoretical solution which is defined as *P2* later, we consider the absolute herd behavior (Wang & Zhao (2024a)). We will study more general optimal investment problems in our future work. Next, to answer **Question B**, we collect real-user data using interviews and questionnaires, and obtain pre-SFT LLMs' investment decisions for *P2*. Then, we show that pre-SFT LLMs' responses are misaligned with the real-user data, and validate the statistical consistency between the theoretical solution and the real-user data.

### 3.1 OPTIMAL INVESTMENT PROBLEMS UNDER HERD BEHAVIOR

Following the prior work in Merton (1969), we consider the scenario where an investor and an investment assistant invest in the period $\mathcal{T}$ in a financial market consisting of a deposit and a stock. We define the funds invested in the stock by the investor and investment assistant as their *investment decisions*, denoted by $\{P(t)\}_{t \in \mathcal{T}}$ and $\{Q(t)\}_{t \in \mathcal{T}}$, respectively. We denote $r$ as the interest rate of the deposit, $v$ and $\sigma$ as the excess return rate and volatility of the stock, and $T$ as the terminal time. Given the above parameters, the investor's fund $\{X(t)\}_{t \in \mathcal{T}}$ satisfies

$$\mathrm{d}X(t) = [rX(t) + vP(t)]\mathrm{d}t + \sigma P(t)\mathrm{d}W(t),\ t \in \mathcal{T}, \tag{1}$$

where $X(0) = x_0$ is his/her initial fund, and $\{W(t)\}_{t \in \mathcal{T}}$ is a standard Brownian motion modeling the randomness of the stock price. We assume that the investment assistant is rational and tries to maximize his/her expected utility of the terminal wealth, and from Rogers (2013), we assume that the investment assistant's decision $\{Q(t)\}_{t \in \mathcal{T}}$ satisfies $Q(t) = \frac{v}{A\sigma^2}\exp[r(t-T)], t \in \mathcal{T}$, where $A$ is the investment assistant's risk aversion coefficient (Pratt (1978)).

Herd behavior can be categorized into two types: (1) absolute herd behavior, where investors replicate the entire portfolio of the investment assistant; (2) relative herd behavior, where they adjust their investments in response to changes made by the investment assistant within the same time interval and imitate the changing rate of the investment assistant's decision (Lakonishok et al. (1992); Wang & Zhao (2024b)). Considering the herd behavior, the investor jointly maximizes his/her expected utility of the terminal fund $\mathbb{E}\phi[X(T)]$ and minimizes the distance between his/her own and the investment assistant's decisions $D(P, Q)$. Following the prior work in Rogers (2013), we assume that the investor's utility of the terminal fund is $\phi[X(T)] = -\frac{1}{\alpha}\exp[-\alpha X(T)]$, where $\alpha$ is his/her risk aversion coefficient. In summary, the optimal investment problem under herd behavior is

$$\sup_{\{P(t)\}_{t \in \mathcal{T}}} \mathbb{E}\phi[X(T)] - \theta D(P, Q), \tag{2}$$

where $\theta$ is the influence coefficient to address the tradeoff between the two different objectives. We call the risk aversion coefficient $\alpha$ and the influence coefficient $\theta$ the investor's *investment attribute*.

***P1*: Optimal investment problem under relative herd behavior.** Following the prior work in Wang & Zhao (2024b), when considering the relative herd behavior, the distance is defined as $\delta(P, Q) = \frac{1}{2}\int_0^T [P'(t) - Q'(t)]^2 \mathrm{d}t$, i.e., the integrated square error between the two decisions' changing rates. In this case, the optimal investment problem is

$$\textit{\textbf{P1}}:\ \sup_{\{P(t)\}_{t \in \mathcal{T}}} \mathbb{E}\phi[X(T)] - \theta\delta(P, Q). \tag{3}$$

To ensure that *P1* has a unique solution, we must add two boundary conditions: $P'(0) = Q'(0)$ and $P'(T) = Q'(T)$, which represents that the investor's decision-changing rates at the initial and terminal times are equal to those of the investment assistant (Wang & Zhao (2024b)).

*P1* has a theoretical solution which is complex to compute (Wang & Zhao (2024b)). We consider a similar and simpler problem, i.e., the optimal investment problem under absolute herd behavior.

***P2*: Optimal investment problem under absolute herd behavior.** For the case of the absolute herd behavior, the distance is defined as $\Delta(P, Q) = \frac{1}{2}\int_0^T [P(t) - Q(t)]^2 \mathrm{d}t$, i.e., the integrated square error between the two decisions (Wang & Zhao (2024a)), and the optimal investment problem is

$$\textit{\textbf{P2}}:\ \sup_{\{P(t)\}_{t \in \mathcal{T}}} \mathbb{E}\phi[X(T)] - \theta\Delta(P, Q). \tag{4}$$

From the work in Wang & Zhao (2024a), the theoretical optimal decision for *P2* is

$$\hat{P}(t) = \frac{A\sigma^2\eta\exp[2r(T-t)]+\theta}{\alpha\sigma^2\eta\exp[2r(T-t)]+\theta} \cdot \frac{v}{A\sigma^2}\exp[r(t-T)],\ t \in \mathcal{T}, \tag{5}$$

where the parameter $\eta$ can be numerically calculated using Algorithm 1 in Appendix A.1.

We set the parameter values in *P1* and *P2* according to Wang & Zhao (2024a) and Wang & Zhao (2024b), as shown in Appendix A.2.

### 3.2 DATA COLLECTION

**Real-User Data Collection.** To verify whether the theoretical solution in equation 5 matches users' real investment decisions, we collect real-user data from 119 participants using interviews and questionnaires when facing the investment problem *P2*. We denote the index set of participants as

$\mathcal{I} = \{1, 2, \ldots, 119\}$. To reduce bias and noise in the collected data, we primarily recruit professionals and students in the fields of economics and finance, and we treat this real-user data as a proxy for the ground truth.

The questionnaire we use is in Figure 6 in Appendix A.7. In the first part, we provide the task description, including information on the deposit and stock as well as the participants' goals. In the second part, participants report their investment decisions, denoted by $\{\tilde{P}_i(t)\}_{t\in\mathcal{T}}$ for all $i \in \mathcal{I}$. To facilitate participants' decision-making, we ask them to report the proportions of funds invested in the stock, i.e., $\{\tilde{P}_i(t)/X_i(t)\}_{t\in\mathcal{T}}$. When processing the data, we first calculate $\{X_i(t)\}_{t\in\mathcal{T}}$ using equation 1, and then calculate the participants' investment decisions $\{\tilde{P}_i(t)\}_{t\in\mathcal{T}}$ according to the proportions $\{\tilde{P}_i(t)/X_i(t)\}_{t\in\mathcal{T}}$.

In the third part, we ask the participants the information about their investment attributes, based on which, we calculate their risk aversion coefficients $\{\alpha_i\}_{i\in\mathcal{I}}$ and influence coefficients $\{\theta_i\}_{i\in\mathcal{I}}$ as follows. From the work in Pratt (1978), the risk aversion coefficient $\alpha_i$ reflects the participant's preference between risky and risk-free options. If the participant is indifferent between the following two options: (1) receiving $w_1$ with probability $p_i$, and receiving nothing with probability $1 - p_i$, or (2) receiving $w_2$, his/her risk aversion coefficient $\alpha_i$ can be determined by solving the equation $p_i = \frac{\exp(\alpha_i w_2)-1}{\exp(\alpha_i w_1)-1}$. We ask the participant to provide his/her response for $p_i$, from which we calculate his/her risk aversion coefficient $\alpha_i$. The influence coefficient $\theta$ quantifies the level of herd behavior. In the third part of the questionnaire, we ask participants: "On a scale from 0 to 10, how much do you rely on the investment assistant when making decisions, where 10 represents the highest level of reliance and 0 the lowest?" From the work in Wang & Zhao (2024a), the influence coefficient $\theta_i$ typically falls within the range $[0, 1 \times 10^{-7}]$. Therefore, we calculate the participant's influence coefficient as $\theta_i = k_i \times 10^{-8}$, where $k_i$ is his/her response.

**Collection of Pre-SFT LLMs' Investment Decisions.** Next, to verify whether pre-SFT LLMs align with real-user data, we collect the pre-SFT LLMs' investment decisions. In this work, we choose a variety of LLMs, including API-based model `GPT-3.5-Turbo` (Achiam et al. (2023)), as well as open-source models like `GLM-4-9B-CHAT` (GLM et al. (2024)), `Qwen2-7B-Instruct` (Yang et al. (2024)), and `Meta-Llama-3.1-8B-Instruct` (Dubey et al. (2024)). To obtain these pre-SFT LLMs' investment decisions in *P2*, we first construct a prompt, as shown in Figure 7 in Appendix A.7. The first part is identical to the questionnaire in Figure 6, where we designate the pre-SFT LLM as an investment expert and describe the task. In the second part, we assign the pre-SFT LLM its investment attribute, corresponding to the participant's investment attribute $\{\alpha_i\}_{i\in\mathcal{I}}$ and $\{\theta_i\}_{i\in\mathcal{I}}$ in the real-user data. In the third part, the pre-SFT LLM reports the proportion of its funds invested in the stock $\{P_i(t)/X_i(t)\}_{t\in\mathcal{T}}$. We then obtain the pre-SFT LLM's investment decision $\{P_i(t)\}_{t\in\mathcal{T}}$.

## 3.3 VALIDATION OF PRE-SFT LLMS AND THE THEORETICAL SOLUTION

The real-user data shows that the participants' risk aversion coefficients $\{\alpha_i\}_{i\in\mathcal{I}}$ and influence coefficients $\{\theta_i\}_{i\in\mathcal{I}}$ fall within the ranges of $\tilde{\mathcal{S}}_\alpha = [0.09, 0.38]$ and $\tilde{\mathcal{S}}_\theta = [0, 1 \times 10^{-7}]$, respectively. For the convenience of data processing, we discretize these two sets into $\tilde{\mathcal{S}}_\alpha = \bigcup_{m\in\mathcal{M}} \tilde{\mathcal{S}}_\alpha^m$ and $\tilde{\mathcal{S}}_\theta = \bigcup_{n\in\mathcal{N}} \tilde{\mathcal{S}}_\theta^n$, and treat values that fall within the same interval as the same value[1]. We then group the participants according to these subsets, with participants sharing the same investment attributes forming a class. Specifically, the class of participants with risk aversion coefficient $\alpha \in \tilde{\mathcal{S}}_\alpha^m$ and influence coefficient $\theta \in \tilde{\mathcal{S}}_\theta^n$ for all $m \in \mathcal{M}$ and $n \in \mathcal{N}$ is denoted as $\mathcal{I}^{mn} = \{i | \alpha_i \in \tilde{\mathcal{S}}_\alpha^m, \theta_i \in \tilde{\mathcal{S}}_\theta^n\}$ for all $m \in \mathcal{M}$ and $n \in \mathcal{N}$.

For each participant class $\mathcal{I}^{mn}$, we calculate the mean and the 95% confidence interval of the real-user data, the mean and the 95% confidence interval of the pre-SFT LLMs' investment decisions based on 10 repeated trials with the same investment attribute, and the corresponding theoretical solution. Here, we take the investment attribute $\alpha = 0.13$ and $\theta = 7 \times 10^{-8}$ as an example,

---

[1]Specifically, we set $\tilde{\mathcal{S}}_\alpha = [0.09, 0.13) \cup [0.13, 0.19) \cup [0.19, 0.26) \cup [0.26, 0.38) \cup \{0.38\}$ and $\tilde{\mathcal{S}}_\theta = [0, 1 \times 10^{-8}) \cup [1 \times 10^{-8}, 2 \times 10^{-8}) \cup [2 \times 10^{-8}, 3 \times 10^{-8}) \cup [3 \times 10^{-8}, 4 \times 10^{-8}) \cup [4 \times 10^{-8}, 5 \times 10^{-8}) \cup [5 \times 10^{-8}, 6 \times 10^{-8}) \cup [6 \times 10^{-8}, 7 \times 10^{-8}) \cup [7 \times 10^{-8}, 8 \times 10^{-8}) \cup [8 \times 10^{-8}, 9 \times 10^{-8}) \cup [9 \times 10^{-8}, 1 \times 10^{-7}) \cup \{1 \times 10^{-7}\}$. In the following, we use the left point's value to approximate the entire interval.

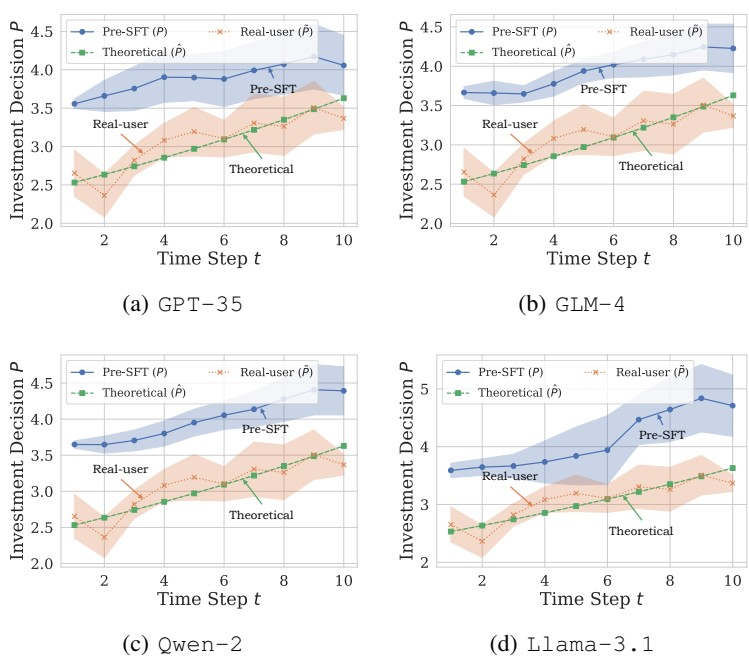

Figure 2: Comparison of real-user data ($\tilde{P}$), pre-SFT LLMs' investment decision ($P$), and theoretical solution ($\hat{P}$) on **P2**.

and observe the same trend for other values. The experimental results are in Figure 2. As shown in Figure 2, there is a significant discrepancy between the pre-SFT LLMs' investment decisions and the real-user data, indicating that pre-SFT LLMs fail to align with real-user data in optimal investment under absolute herd behavior. We also find that the performance of pre-SFT LLMs in optimal investment under relative herd behavior is misaligned, as shown in Figure 5 in Appendix A.3. This underscores the necessity of supervised fine-tuning to bridge the gap between pre-SFT LLMs' investment decisions and real-user data.

On the contrary, from Figure 2, the theoretical solutions are much closer to the real-user data than pre-SFT LLMs' investment decisions. We further employ statistical methods to validate the consistency between the theoretical solutions and real-user data. For the $i$-th participant, we denote his/her real investment decision as $\{\tilde{P}_i(t)\}_{t\in\mathcal{T}}$, and the theoretical solution with the same investment attribute as $\{\hat{P}_i(t)\}_{t\in\mathcal{T}}$, respectively. We first calculate the difference and correlation coefficient between $\{\tilde{P}_i(t)\}_{t\in\mathcal{T}}$ and $\{\hat{P}_i(t)\}_{t\in\mathcal{T}}$, which are defined as

$$d(\tilde{P}_i, \hat{P}_i) = \sum_{t\in\mathcal{T}}[\tilde{P}_i(t) - \hat{P}_i(t)] \text{ and } \rho(\tilde{P}_i, \hat{P}_i) = \frac{\sum_{t\in\mathcal{T}}[\tilde{P}_i(t)-\bar{\tilde{P}}_i][\hat{P}_i(t)-\bar{\hat{P}}_i]}{\sqrt{\sum_{t\in\mathcal{T}}[\tilde{P}_i(t)-\bar{\tilde{P}}_i]^2 \sum_{t\in\mathcal{T}}[\hat{P}_i(t)-\bar{\hat{P}}_i]^2}}, \quad (6)$$

respectively, where $\bar{\tilde{P}}_i = \frac{1}{T}\sum_{t\in\mathcal{T}}\tilde{P}_i(t)$ and $\bar{\hat{P}}_i = \frac{1}{T}\sum_{t\in\mathcal{T}}\hat{P}_i(t)$ are the averages of the $i$-th participant's investment decisions and the theoretical solution at different time steps, respectively. Next, we conduct $t$-tests on the means of the differences $\{d(\tilde{P}_i, \hat{P}_i)\}_{i\in\mathcal{I}}$ and the correlation coefficients $\{\rho(\tilde{P}_i, \hat{P}_i)\}_{i\in\mathcal{I}}$ (Shao (2008)), respectively. For the differences $\{d(\tilde{P}_i, \hat{P}_i)\}_{i\in\mathcal{I}}$, the results show that their mean does not significantly deviate from $0$ at the $1\%$ significance level, with a $t$-statistic $= -1.075$. For the correlation coefficients $\{\rho(\tilde{P}_i, \hat{P}_i)\}_{i\in\mathcal{I}}$, the results show that their mean does not significantly deviate from $0.85$ at the $1\%$ significance level, with a $t$-statistic $= -0.843$. Since a mean difference close to $0$ indicates minimal discrepancy and a correlation coefficient close to $0.85$ reflects a strong positive relationship, we show that there exists significant consistency between the theoretical solution and real-user data.

In summary, due to the significant gap between pre-SFT LLMs and real-user data, fine-tuning the LLMs with the theoretical solution is critical. As the theoretical solution closely aligns with real-user data, we can use them to construct the SFT training dataset as a substitute for real-user data.

# 4 METHODOLOGY: INVESTALIGN

As mentioned in Section 3, we statistically demonstrate the consistency between the theoretical solution and the real-user data. Given this observation, we propose **InvestAlign**, which uses the theoretical solution to efficiently and cost-effectively generate SFT training datasets to fine-tune LLMs to align with real-user data. In this section, to answer **Question C**, i.e., how we can construct the SFT training dataset using the theoretical solution, and whether this training dataset performs better in fine-tuning compared to real-user data, we first introduce the method of constructing SFT training datasets using the theoretical solution. Then, we theoretically prove that training LLMs on these datasets results in faster parameter convergence compared to using real-user data.

## 4.1 CONSTRUCTING SFT TRAINING DATASET WITH THEORETICAL SOLUTION

The SFT training dataset comprises input-output pairs used for fine-tuning LLMs, which are generated based on a custom prompt template. The prompt for SFT is in Figure 8 in Appendix A.7. When constructing the SFT training dataset, we need to vary the investment attribute, i.e., the risk aversion coefficient $\alpha$ and the influence coefficient $\theta$. Following the work in Wang & Zhao (2024a), we set the values of $\alpha$ and $\theta$ in $\hat{S}_\alpha = \{0.05, 0.10, \ldots, 0.50\}$ and $\hat{S}_\theta = \{1 \times 10^{-8}, 2 \times 10^{-8}, \ldots, 1 \times 10^{-7}\}$, respectively. Using the same method in Section 3.2, we set the above investment attributes through two questions expressed in natural language that are easy for LLMs to understand, rather than directly telling them the specific values of these parameters. For each investment attribute, we first calculate the theoretical optimal decision $\{\hat{P}(t)\}_{t \in \mathcal{T}}$ using equation 5 and Algorithm 1, and then calculate the investment proportion $\{\hat{P}(t)/X(t)\}_{t \in \mathcal{T}}$ using equation 1. Note that there exists a random perturbation $\{W(t)\}_{t \in \mathcal{T}}$ in equation 1, and we repeat 10 trials for each investment attribute. In summary, the SFT training dataset contains $10 \times 10 \times 10 = 1000$ training samples.

## 4.2 ANALYSIS OF THE PARAMETER CONVERGENCE RATE IN FINE-TUNING

We theoretically show that fine-tuning LLMs on the training datasets constructed from theoretical solutions leads to faster parameter convergence compared to using real-user data.

To gain insights and ensure mathematical tractability, we make the following assumptions. First, when calculating the loss function, we only consider the values of the LLM's investment decision, theoretical solution, and real-user data, excluding the natural language parts. This is because the natural language parts for all three experiments are the same. Second, we assume that the sample size of the training dataset constructed from the theoretical solution and real-user data are both sufficiently large. Third, we assume that the output layer of the LLM is a Sigmoid layer, i.e., $\text{Sigmoid}(\mathbf{z}) = \frac{1}{1+\exp(-\mathbf{w}^\top \mathbf{z})}$, where $\mathbf{w}$ is the model parameter and $\mathbf{z}$ is the output layer's input. Although the output layer of the LLM may be more complex, this simplification makes the theoretical analysis tractable. We denote the ranges of the LLM's investment decision $P(t)$, theoretical solution $\hat{P}(t)$, and real-user data $\tilde{P}(t)$ as $\mathcal{P}(t)$, $\hat{\mathcal{P}}(t)$, and $\tilde{\mathcal{P}}(t)$, respectively.

Given the above assumptions, in the following, we analyze the parameter convergence rate in fine-tuning. First, according to the second assumption, when fine-tuning the LLM using the training dataset constructed from the theoretical solution, we can express the cross-entropy loss function as

$$\hat{L}(\mathbf{w}) = -\sum_{t \in \mathcal{T}} \int_{\hat{\mathcal{P}}(t)} f_{\hat{P}(t)}(x) \log f_{P(t)}(x) \mathrm{d}x, \tag{7}$$

where $f_{P(t)}(x)$ and $f_{\hat{P}(t)}(x)$ represent the probability density functions of $P(t)$ and $\hat{P}(t)$ in the training dataset, respectively. Similarly, we can define the cross-entropy loss function $\tilde{L}(\mathbf{w})$ for the case when fine-tuning the LLM using the real-user data.

Next, we derive the analytical form of $f_{\hat{P}(t)}(x)$ and $f_{\tilde{P}(t)}(x)$. When we construct the SFT training dataset, we uniformly set the values of the risk aversion coefficient $\alpha$ and the influence coefficient $\theta$ within a rectangular region. Therefore, we assume that $\alpha$ and $\theta$ satisfy two uniform distributions. As shown in in Appendix A.4, we can prove that the theoretical optimal decisions $\{\hat{P}(t)\}_{t \in \mathcal{T}}$ approximately satisfies a Pareto distribution, i.e.,

$$f_{\hat{P}(t)}(x) \approx \frac{c}{x^2}, \ x \in \hat{\mathcal{P}}(t), \tag{8}$$

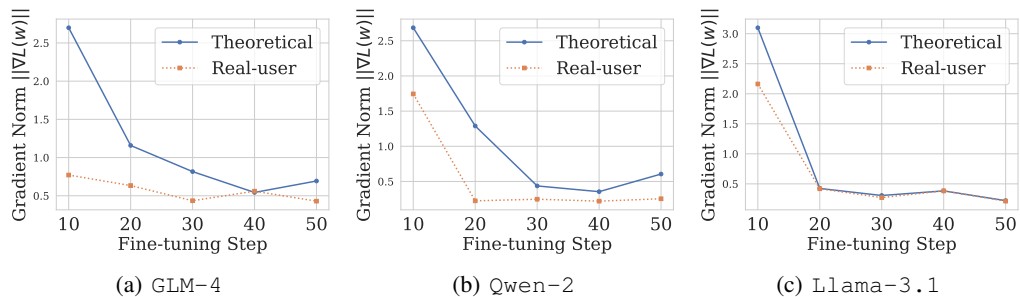

Figure 3: Comparison of the gradient norms between using theoretical solution and real-user data.

where $c$ is the normalization parameter. Equation 10 is consistent with the empirical research in the field of finance, which shows that the distribution of investor trading volume often exhibits a power-law characteristic (Iori (2002)). From equation 10, we can find that the probability distribution function $f_{\hat{P}(t)}(x)$ is a monotonically decreasing function. Thus, we assume that $f_{P(t)}(x)$ is also monotonically decreasing. Because the real-user data often have a bigger noise than the theoretical solution, we assume that $\tilde{P}(t)$ is $\hat{P}(t)$ plus a white noise $n(t)$, i.e, $\tilde{P}(t) = \hat{P}(t) + n(t)$, where $\{n(t)\}_{t \in \mathcal{T}}$ are independent and identically satisfy a uniform distribution $U(-\varepsilon, \varepsilon)$. Using the convolution formula (Rényi (2007)), we have

$$f_{\tilde{P}(t)}(x) \approx \frac{c}{2\varepsilon} \left( \frac{1}{\max\{\min[\hat{\mathcal{P}}(t)], x-\varepsilon\}} - \frac{1}{\min\{\max[\hat{\mathcal{P}}(t)], x+\varepsilon\}} \right), \, x \in \tilde{\mathcal{P}}(t), \quad (9)$$

Finally, given $f_{\hat{P}(t)}(x)$ and $f_{\tilde{P}(t)}(x)$, we calculate the gradient norms of the loss function, which are

$$\|\nabla \hat{L}(\mathbf{w})\| = \|\mathbf{z}\| \sum_{t \in \mathcal{T}} \left[ 1 - \int_{\hat{\mathcal{P}}(t)} f_{\hat{P}(t)}(x) f_{P(t)}(x) \mathrm{d}x \right], \text{ and} \quad (10)$$

$$\|\nabla \tilde{L}(\mathbf{w})\| = \|\mathbf{z}\| \sum_{t \in \mathcal{T}} \left[ 1 - \int_{\tilde{\mathcal{P}}(t)} f_{\tilde{P}(t)}(x) f_{P(t)}(x) \mathrm{d}x \right], \quad (11)$$

respectively. From equations 8 – 11, we can further prove that

$$\|\nabla \hat{L}(\mathbf{w})\| > \|\nabla \tilde{L}(\mathbf{w})\|. \quad (12)$$

Details are in Appendix A.5. That is, the gradient norm when using the training dataset constructed from the theoretical solution is higher than when using real-user data. This is because, once the parameters are given, the real-user data are noisy, while the theoretical solution is deterministic. According to Chen & Yang (2018), the gradient descent algorithm converges faster when the gradient norm is larger. Thus, from equation 12, we can draw the conclusion that the gradient descent algorithm converges faster when using the training dataset compared to using real-user data.

We conduct an experiment to validate our above analysis on open-source models including `GLM-4-9B-CHAT`, `Qwen2-7B-Instruct`, and `Llama-3.1-8B-Instruct`. We construct the SFT training datasets using both the theoretical solution and real-user data, and fine-tune the LLMs with these training datasets using low-rank adaptation (LoRA) in Hu et al. (2021). We set the LoRA rank, alpha, and dropout rate as $4$, $32$, and $0.1$, respectively, and keep the training parameters, such as the learning rate and batch size, etc., unchanged. The experimental results of the gradient norm $\|\nabla L(\mathbf{w})\|$ are in Figure 3. From Figure 3, the gradient norm when using the training dataset constructed from theoretical solution is significantly higher than when using real-user data across different LLMs, validating that fine-tuning LLMs on the training datasets constructed from theoretical solution leads to faster parameter convergence compared to using real-user data.

## 5 EXPERIMENTS AND PERFORMANCE VALIDATION

In this section, to answer **Question D**, we conduct experiments to verify the alignment performance of **InvestAgent**s with real-user data in the simple problem *P2* and the original problem *P1*.

### 5.1 ALIGNMENT PERFORMANCE OF **INVESTAGENT** IN *P2*

**Experimental Setup.** To compare the alignment performance of pre-SFT LLMs and **InvestAgent**s with real-user data, we develop a Python-based simulation environment where the investor's fund is

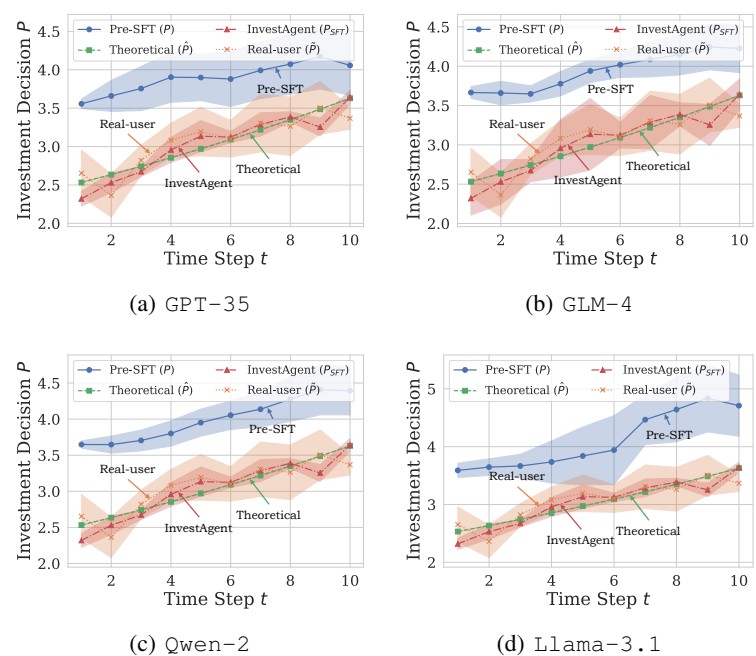

Figure 4: Comparison of real-user data ($\tilde{P}$), pre-SFT LLMs' investment decision ($P$), **InvestAgent**s' investment decision ($P_{SFT}$), and theoretical solution ($\hat{P}$).

updated according to equation 1. The prompt used in the experiment is in Figure 7 in Appendix A.7. For different investment attributes, we select $\alpha$ from $\mathcal{S}_{\alpha} = \{0.09, 0.13, 0.19, 0.26, 0.38\}$ and $\theta$ from $\mathcal{S}_{\theta} = \{0, 1 \times 10^{-8}, \ldots, 1 \times 10^{-7}\}$. Given the random perturbation $\{W(t)\}_{t \in \mathcal{T}}$ in equation 1, we use 10 random seeds for each investment attribute, producing a total of $10 \times 5 \times 11 = 550$ trials.

**Experimental Results.** Similarly to the data-processing method in Section 3.3, we plot the mean and the 95% confidence interval of the real-user data, denoted by $\tilde{P}$, and the pre-SFT LLMs' and **InvestAgent**s' investment decisions based on 10 repeated trials with the corresponding investment attribute, denoted by $P$, and $P_{SFT}$, respectively. We also plot the theoretical solutions, denoted by $\hat{P}$. The experimental results are in Figure 4. Here, we take the investment attribute $\alpha = 0.13$ and $\theta = 7 \times 10^{-8}$ as an example, and we observe the same trend for other values. As shown in Figure 4, **InvestAgent**s' investment decisions are significantly closer to real-user data and theoretical solutions compared to pre-SFT LLMs across different LLMs.

Additionally, to quantitatively evaluate how **InvestAlign** can help pre-SFT LLMs align with real-user data in *P2*, we calculate the overall MSE between the mean of pre-SFT LLMs' investment decisions and real-user data, which is

$$\text{Overall MSE}(P, \tilde{P}) = \frac{1}{|\mathcal{M}||\mathcal{N}||\mathcal{T}|} \sum_{m \in \mathcal{M}} \sum_{n \in \mathcal{N}} \sum_{t \in \mathcal{T}} [P_{mn}(t) - \tilde{P}_{mn}(t)]^2, \quad (13)$$

and the overall MSE between the mean of **InvestAgent**s' investment decisions and real-user data:

$$\text{Overall MSE}(P_{SFT}, \tilde{P}) = \frac{1}{|\mathcal{M}||\mathcal{N}||\mathcal{T}|} \sum_{m \in \mathcal{M}} \sum_{n \in \mathcal{N}} \sum_{t \in \mathcal{T}} [P_{SFT,mn}(t) - \tilde{P}_{mn}(t)]^2, \quad (14)$$

where $\{\tilde{P}_{mn}(t)\}_{t \in \mathcal{T}}$ represents the mean of the real-user data in class $\mathcal{I}^{mn}$, $\{P_{mn}(t)\}_{t \in \mathcal{T}}$ and $\{P_{SFT,mn}(t)\}_{t \in \mathcal{T}}$ represents the mean of the pre-SFT LLMs' and **InvestAgent**s' investment decisions with the corresponding investment attribute, respectively. The experimental results are in Table 1. As shown in Table 1, **InvestAlign** helps reduce the overall MSEs by $45.59\% \sim 61.26\%$.

Furthermore, we also conduct an ablation study on the hyper-parameters of fine-tuning, including LoRA Rank and fine-tuning steps, as shown in Appendix A.6. We find that the overall MSE decreases as either LoRA Rank or fine-tuning steps increase.

The experimental results validate the effectiveness of our proposed method **InvestAlign**, i.e., fine-tuning LLMs using the SFT training dataset constructed from the theoretical solution can align them better with investor decision-making under herd behavior.

Table 1: Comparison of the overall MSE between pre-SFT LLMs' and **InvestAgent**s' investment decisions with real-user data in optimal investment problems *P2* and *P1*.

| Overall MSE | GPT-35 | GLM-4 | Qwen-2 | Llama-3.1 |
|---|---|---|---|---|
| ***P2*: Absolute herd behavior** | | | | |
| **Pre-SFT LLM** | 4.44 | 4.20 | 3.97 | 4.08 |
| **InvestAgent** | 1.72 | 2.26 | 2.16 | 1.59 |
| **Reduction from Pre-SFT (%)** | -61.26% | -46.19% | -45.59% | -61.03% |
| ***P1*: Relative herd behavior** | | | | |
| **Pre-SFT LLM** | 14.03 | 13.85 | 17.22 | 13.07 |
| **InvestAgent** | 7.46 | 6.14 | 7.46 | 7.25 |
| **Reduction from Pre-SFT (%)** | -46.84% | -55.66% | -56.69% | -44.52% |

## 5.2 PERFORMANCE OF **INVESTAGENT** IN *P1*

**Experimental Setup.** This experiment shows the alignment performance of our proposed **InvestAlign**, i.e., using LLMs fine-tuned from *P2* to solve *P1*. The prompt we use is in Figure 9 in Appendix A.7. The investment attributes are set the same as those in Section 5.1. We collect 90 real-user data using interviews and questionnaires, and the participants are also primarily professionals and students in the fields of economics and finance to reduce bias and noise in collected data.

**Experimental Results.** Using the same method in Section 5.1, we plot the overall MSE between the mean of pre-SFT LLMs' investment decisions with real-user data, Overall MSE$(P, \tilde{P})$, and the overall MSE between the mean of **InvestAgent**s' investment decisions with real-user data, Overall MSE$(P_{SFT}, \tilde{P})$, in Table 1. As shown in Table 1, **InvestAlign** helps reduce the overall MSEs by $44.53\% \sim 56.68\%$. The experiment results validate the effectiveness of our proposed **InvestAlign**, and show that the **InvestAgent**s fine-tuned using the theoretical solution in a similar and simpler problem can better align with human decision-making processes in a complex problem than pre-SFT LLMs. It demonstrates the potential of **InvestAlign** to solve complex optimal investment problems and align LLMs with investor decision-making processes in economics and finance.

In addition to the experiments mentioned above, we also: 1) supplement smaller samples of real-user data with theoretical solutions to construct a training dataset to improve robustness; 2) compare **InvestAgent**s with LLMs fine-tuned using the baseline FinGPT dataset (Yang et al. (2023a)). The experimental results and analysis are in Appendix A.8 and Appendix A.9, respectively.

## 6 CONCLUSION

Studying investor decision-making processes under the herd behavior is of great significance in microeconomics and behavioral finance. LLMs can be leveraged to assist in solving complex investment problems. To fine-tune LLMs for alignment with human decision-making processes, a substantial amount of real-user data is required. However, the cost of collecting the real-user data is high, and there are concerns regarding privacy and security. To address these challenges, we propose **InvestAlign**, a novel method that constructs training datasets using the theoretical solution of a similar and simple problem to align LLMs with investor behavior under herd behavior. We demonstrate that fine-tuning LLMs on these training datasets leads to faster parameter convergence compared to using real-user data. The experimental results indicate that **InvestAgent**s, fine-tuned with **InvestAlign**, achieves superior alignment performance in the original complex problem.

As an initial exploration in this field, **InvestAlign** does not claim universal applicability to all complex optimal investment problems, and its theoretical solutions may not fully encapsulate the intricacies of real-world investor behavior. Our primary focus is on addressing a specific challenge: data scarcity in training LLMs for investor decision-making. In future work, we plan to explore its applicability across diverse investor profiles and complex behavioral biases. Additionally, we aim to investigate the impact of RLHF on **InvestAgent**s and compare it with SFT to assess the effectiveness of different alignment strategies in investment decision-making tasks.

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

# A  APPENDIX

## A.1  THEORETICAL OPTIMAL DECISIONS OF *P2*

The investor's optimal decision for *P2* is

$$\hat{P}(t) = \frac{A\sigma^2\eta \exp[2r(T-t)]+\theta}{\alpha\sigma^2\eta \exp[2r(T-t)]+\theta} \cdot \frac{v}{A\sigma^2} \exp[r(t-T)], \ t \in \mathcal{T}, \tag{15}$$

where the parameter $\eta$ can be numerically calculated using Algorithm 1. The proof can be found in Wang & Zhao (2024a).

---

**Algorithm 1:** Numerical Method of the Parameter $\eta$ in *P2*.

---

**Input:** Interest rate: $r$;
  Excess return rate: $v$;
  Volatility: $\sigma$;
  Initial fund: $x_0$;
  Risk aversion coefficients: $\alpha$ and $A$;
  Investment period: $T$;
  Influence coefficient: $\theta$;
  Tolerance: $\varepsilon$.

**Output:** The parameter $\eta$.

$\eta_0 = \exp\left[-\alpha x_0 e^{rT} - \frac{v^2 T}{2\sigma^2}\right], \Delta\eta_0 = +\infty, k = 0, \vartheta = \frac{\theta}{\alpha\sigma^2}$;

**while** $\Delta\eta_k \geqslant \varepsilon$ **do**

$\quad \eta_{k+1} = \eta_0 \exp\left\{\int_0^T \frac{\vartheta^2 v^2 (\alpha/A-1)^2 \mathrm{d}t}{2\sigma^2 \left\{\eta_k e^{2r(T-t)}+\vartheta\right\}^2}\right\}$;

$\quad \Delta\eta_{k+1} = |\eta_{k+1} - \eta_k|$;

$\quad k \leftarrow k + 1$;

**end**

$\eta \approx \eta_k$.

---

## A.2  PARAMETER SETTING

Following the prior work in Wang & Zhao (2024a), we set the parameter values as follows.

- Interest rate: $r = 0.04$;
- Excess return rate: $v = 0.03$;
- Volatility: $\sigma = 0.17$;
- Initial fund: $x_0 = 10$;
- Investment assistant's risk aversion coefficient: $A = 0.02$;
- Investment period: $T = 10$.

### A.3 COMPARISON OF REAL-USER DATA AND PRE-SFT LLMS' INVESTMENT DECISION ON *P1*

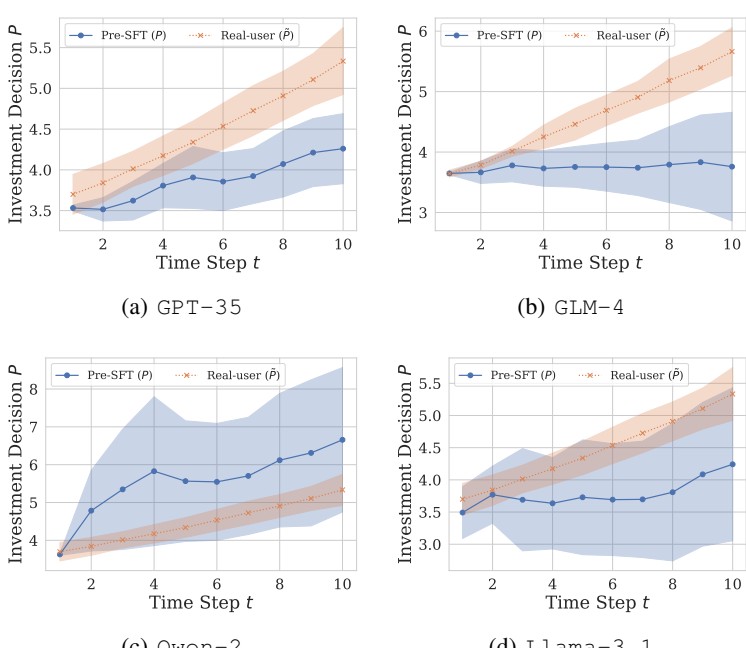

(a) `GPT-35`

(b) `GLM-4`

(c) `Qwen-2`

(d) `Llama-3.1`

Figure 5: Comparison of real-user data ($\tilde{P}$) and pre-SFT LLMs' investment decision ($P$) on *P1*.

### A.4 PROBABILITY DISTRIBUTION FUNCTION OF THE OPTIMAL DECISION

We assume the parameters $\alpha$ and $\theta$ satisfy two uniform distributions, denoted by $U(\underline{\alpha}, \overline{\alpha})$ and $U(\underline{\theta}, \overline{\theta})$, respectively. Therefore, their probability distribution functions are

$$f_\alpha(x) = \tfrac{1}{\overline{\alpha}-\underline{\alpha}},\ x \in [\underline{\alpha}, \overline{\alpha}],\ \text{and}\ f_\theta(x) = \tfrac{1}{\overline{\theta}-\underline{\theta}},\ x \in [\underline{\theta}, \overline{\theta}]. \tag{16}$$

From equation 5, using the convolution formula (Rényi (2007)), we have

$$f_{\hat{P}(t)}(x) = \tfrac{1}{\overline{\theta}-\underline{\theta}} \int_{\underline{\theta}}^{\overline{\theta}} f_\alpha \left( \tfrac{1}{\sigma^2 \eta \mathrm{e}^{2r(T-t)}} \left[ \tfrac{A\sigma^2 \eta \mathrm{e}^{2r(T-t)}+y}{x} \cdot \tfrac{v}{A\sigma^2}\mathrm{e}^{r(t-T)} - y \right] \right)$$
$$\cdot \tfrac{A\sigma^2 \eta \mathrm{e}^{2r(T-t)}+y}{\sigma^2 \eta \mathrm{e}^{2r(T-t)} x^2} \cdot \tfrac{v}{A\sigma^2}\mathrm{e}^{r(t-T)}\mathrm{d}y. \tag{17}$$

Here, following the prior work in Wang & Zhao (2024b), we assume that $\eta$ remains constant when $\alpha$ and $\theta$ change slightly. Because $\hat{P}(t) \in \hat{\mathcal{P}}(t)$, we can rewrite equation 17 as

$$f_{\hat{P}(t)}(x) \approx \tfrac{\min[\hat{\mathcal{P}}(t)] \cdot \max[\hat{\mathcal{P}}(t)]}{\max[\hat{\mathcal{P}}(t)] - \min[\hat{\mathcal{P}}(t)]} \cdot \tfrac{1}{x^2},\ x \in \hat{\mathcal{P}}(t). \tag{18}$$

That is, the theoretical optimal decision $\hat{P}(t)$ approximately satisfies a Pareto distribution. To simplify the notation, we denote the normalization parameter $c = \tfrac{\min[\hat{\mathcal{P}}(t)] \cdot \max[\hat{\mathcal{P}}(t)]}{\max[\hat{\mathcal{P}}(t)] - \min[\hat{\mathcal{P}}(t)]}$.

## A.5 GRADIENT NORMS OF THE LOSS FUNCTION

First, we prove equation 9 from equation 8. Using the convolution formula (Rényi (2007)), we have

$$
f_{\tilde{P}(t)}(x) \approx \frac{1}{2\varepsilon} \int_{-\varepsilon}^{\varepsilon} f_{\hat{P}(t)}(x - y) \mathrm{d}y
$$

$$
= \begin{cases} \frac{c}{2\varepsilon} \int_{\min[\hat{\mathcal{P}}(t)]-x}^{\varepsilon} \frac{1}{(x-y)^2} \mathrm{d}y, & x \in [\min[\hat{\mathcal{P}}(t)] - \varepsilon, \min[\hat{\mathcal{P}}(t)] + \varepsilon) \\ \frac{c}{2\varepsilon} \int_{-\varepsilon}^{\varepsilon} \frac{1}{(x-y)^2} \mathrm{d}y, & x \in [\min[\hat{\mathcal{P}}(t)] + \varepsilon, \max[\hat{\mathcal{P}}(t)] - \varepsilon) \\ \frac{c}{2\varepsilon} \int_{-\varepsilon}^{\max[\hat{\mathcal{P}}(t)]-x} \frac{1}{(x-y)^2} \mathrm{d}y, & x \in [\max[\hat{\mathcal{P}}(t)] - \varepsilon, \max[\hat{\mathcal{P}}(t)] + \varepsilon] \end{cases}
$$

$$
= \frac{c}{2\varepsilon} \left( \frac{1}{\max\{\min[\hat{\mathcal{P}}(t)], x-\varepsilon\}} - \frac{1}{\min\{\max[\hat{\mathcal{P}}(t)], x+\varepsilon\}} \right), \ x \in \tilde{\mathcal{P}}(t). \tag{19}
$$

Next, we prove equation 10 from equation 7. We have

$$
\nabla \hat{L}(\mathbf{w}) = -\sum_{t \in \mathcal{T}} \int_{\hat{\mathcal{P}}(t)} f_{\hat{P}(t)}(x) \nabla \log f_{P(t)}(x) \mathrm{d}x
$$

$$
= -\sum_{t \in \mathcal{T}} \int_{\hat{\mathcal{P}}(t)} \frac{f_{\hat{P}(t)}(x)}{f_{P(t)}(x)} \nabla \mathrm{Sigmoid}(\mathbf{z}) \mathrm{d}x
$$

$$
= -\mathbf{z} \sum_{t \in \mathcal{T}} \int_{\hat{\mathcal{P}}(t)} f_{\hat{P}(t)}(x)[1 - f_{P(t)}(x)] \mathrm{d}x
$$

$$
= -\mathbf{z} \sum_{t \in \mathcal{T}} \left[ \int_{\hat{\mathcal{P}}(t)} f_{\hat{P}(t)}(x) \mathrm{d}x - \int_{\hat{\mathcal{P}}(t)} f_{\hat{P}(t)}(x) f_{P(t)} \mathrm{d}x \right]
$$

$$
= -\mathbf{z} \sum_{t \in \mathcal{T}} \left[ 1 - \int_{\hat{\mathcal{P}}(t)} f_{\hat{P}(t)}(x) f_{P(t)} \mathrm{d}x \right]. \tag{20}
$$

Therefore, the gradient norm is

$$
\|\nabla \hat{L}(\mathbf{w})\| = \|\mathbf{z}\| \sum_{t \in \mathcal{T}} \left[ 1 - \int_{\hat{\mathcal{P}}(t)} f_{\hat{P}(t)}(x) f_{P(t)}(x) \mathrm{d}x \right]. \tag{21}
$$

Using the same method as above, we can prove equation 11. Next, we compare the two gradient norms $\|\nabla \hat{L}(\mathbf{w})\|$ and $\|\nabla \tilde{L}(\mathbf{w})\|$. From equations 10 and 11, we only need to compare the following two integrals: $\int_{\hat{\mathcal{P}}(t)} f_{\hat{P}(t)}(x) f_{P(t)}(x) \mathrm{d}x$ and $\int_{\tilde{\mathcal{P}}(t)} f_{\tilde{P}(t)}(x) f_{P(t)}(x) \mathrm{d}x$. Because the investment decisions of the LLM without SFT can be arbitrary due to the randomness of model parameters, we have $\hat{\mathcal{P}}(t) \subset \tilde{\mathcal{P}}(t) \subset \mathcal{P}(t)$. Because $f_{P(t)}(x)$ is monotonically decreasing, from equations 8 and 9, we can prove that

$$
\int_{\hat{\mathcal{P}}(t)} f_{\hat{P}(t)}(x) f_{P(t)}(x) \mathrm{d}x < \int_{\tilde{\mathcal{P}}(t)} f_{\tilde{P}(t)}(x) f_{P(t)}(x) \mathrm{d}x < 1, \tag{22}
$$

and thus we have

$$
\|\nabla \hat{L}(\mathbf{w})\| > \|\nabla \tilde{L}(\mathbf{w})\|. \tag{23}
$$

## A.6 ABLATION STUDY ON THE HYPER-PARAMETERS OF SFT

We conduct an ablation study on the hyper-parameters of fine-tuning, including LoRA Rank and fine-tuning steps. Here, we take `Qwen-2` and `Llama-3.1` as examples. The experimental results are in Tables 2 and Table 3. It can be seen that our **InvestAlign** consistently enhances the agreement between the **InvestAgent** and real-user data across various hyperparameters. Furthermore, the overall MSE decreases as the strength of fine-tuning increases, either through a larger LoRA Rank or more fine-tuning steps, underscoring the effectiveness of **InvestAlign**. We hypothesize that full-parameter fine-tuning could yield even better results if computational resources permit, which we plan to explore in future studies.

Table 2: Ablation study on the LoRA rank ($R$) using `Qwen-2` and `Llama-3.1`.

| Overall MSE | Qwen-2 | | | Llama-3.1 | | |
|---|---|---|---|---|---|---|
| | $R = 4$ | $R = 8$ | $R = 16$ | $R = 4$ | $R = 8$ | $R = 16$ |
| **Pre-SFT LLM** | 3.97 | 3.97 | 3.97 | 4.08 | 4.08 | 4.08 |
| **InvestAgent** | 3.09 | 2.16 | 1.35 | 2.40 | 1.59 | 1.36 |
| **Reduction from Pre-SFT (%)** | -22.17% | -45.60% | -65.99% | -41.18% | -61.03% | -66.67% |

Table 3: Ablation study on the fine-tuning step ($S$) using `Qwen-2` and `Llama-3.1`.

| Overall MSE | Qwen-2 | | | | | Llama-3.1 | | | | |
|---|---|---|---|---|---|---|---|---|---|---|
| | $S=50$ | $S=100$ | $S=150$ | $S=200$ | $S=250$ | $S=50$ | $S=100$ | $S=150$ | $S=200$ | $S=250$ |
| **Pre-SFT LLM** | 3.97 | 3.97 | 3.97 | 3.97 | 3.97 | 4.08 | 4.08 | 4.08 | 4.08 | 4.08 |
| **InvestAgent** | 2.83 | 3.01 | 2.67 | 2.43 | 2.16 | 3.17 | 2.72 | 2.92 | 1.97 | 1.59 |
| **Reduction from Pre-SFT (%)** | -28.71% | -24.18% | -32.75% | -38.79% | -45.59% | -22.30% | -33.33% | -28.43% | -51.72% | -61.03% |

## A.7 QUESTIONNAIRE AND PROMPTS

---

**Questionnaire for real-user data in *P2***

**1. Task Description**
Starting from next year, you plan to use a portion of your savings (10 million dollars) to invest in a stock and a deposit as part of your personal retirement fund. You will establish a dedicated account to manage this retirement fund. This means you will make a one-time deposit of 10 million dollars into this account and will not deposit any additional funds or withdraw any funds from this account afterward.
**The annualized return of the stock is 7%, with a volatility of 17%.** An annualized return of 7% means that if you invest $100 in this stock, you can expect to have $107 after one year on average (the original $100 plus $7 in return). A volatility of 17% indicates that:
With a 68% probability, the price will be between $100 \pm $17 (i.e., $83 to $117) after one year.
With a 95% probability, the price will be between $100 \pm 2 \times $17 (i.e., $66 to $134) after one year.
With a 99.7% probability, the price will be between $100 \pm 3 \times $17 (i.e., $49 to $151) after one year.
**The annualized return of the deposit is 4%.** If you invest $100 in the deposit, you will receive $104 after one year (the original $100 plus $4 in return).
Over the next 10 years, you will make investment and savings decisions once per year, for a total of {T} decisions. These 10 decision points are labeled 1, 2, ..., 10. At the beginning of year t ($1 \leq t \leq 10$), let the funds in your dedicated account be X(t). Your decision is to allocate part of these funds to invest in the stock, denoted as P(t); the remaining funds will be allocated to savings, which will be X(t) - P(t). **You will determine the proportion of funds to allocate to the stock, i.e., P(t) / X(t).**
During the decision-making process, we will provide you with a **investment assistant** developed by **Omitted for Anonymity**. The investment assistant will provide you with auxiliary information at each decision point. You can refer to the investment assistant's recommendations to some extent, but note that these recommendations may not be optimal. You should also use your own investment insights to avoid blindly following the investment assistant.
**Your goal is to maximize the total amount of funds after 10 years and minimize the risk.**

**2. Investment Decisions**
Now, you have 10 million dollars for investment and savings, and the investment assistant recommends the following investment proportions for the stock over the 10 years: [36.21%, 35.59%, 34.96%, 34.35%, 33.73%, 33.13%, 32.53%, 31.93%, 31.34%, 30.75%]. Considering the investment assistant's recommendations, based on your own investment insights, what is your decided investment proportion sequence for the stock over these 10 years? You need to give a list containing 10 percentages, with each percentage ranging from 0% to 100% and precise to two decimal places, representing the investment proportion for each year t. For example, [34.79%, 38.58%, 35.75%, 32.17%, 31.61%, 30.52%, 34.01%, 32.48%, 34.20%, 31.70%]. You need to replace this percentage list with your actual investment proportion sequence. [________]

**3. Your Investment Characteristics**
(1) At what probability (denoted by p) are the following two choices indifferent to you? A. A probability p of receiving $20, and a probability 1 - p of receiving nothing. B. Receiving $6. [________]
(2) When making a decision, how much do you rely on the investment assistant? Please directly give an integer between 0 and 10. 10 means you rely heavily on the investment assistant, and 0 means you rely little on him/her. [________]

---

Figure 6: Questionnaire for real-user data in *P2*.

**Prompt for pre-SFT LLMs and InvestAgents in *P2*.**

# Task Description
## Background
Assume you are an investment expert. Starting from next year, you plan to use a portion of your savings (10 million dollars) to invest in (1) a stock (hereinafter referred to as **Investment**) and (2) a deposit (hereinafter referred to as **Savings**) as part of your personal retirement fund. You will establish a dedicated account to manage this retirement fund. This means you will make a one-time deposit of 10 million dollars into this account and will not deposit any additional funds or withdraw any funds from this account afterward. Please remember that you need to provide the proportion of funds allocated to the stock each year over the 10 years in the form of a percentage list, rather than providing decision-making recommendations or writing code.

## Financial Market
**Information on the stock**: **The annualized return of the stock is 7%, with a volatility of 17%.** An annualized return of 7% means that if you invest $100 in this stock, you can expect to have $107 after one year on average (the original $100 plus $7 in return). A volatility of 17% indicates that:
With a 68% probability: The asset price will be between $100 ± $17 (i.e., $83 to $117) after one year.
With a 95% probability: The asset price will be between $100 ± 2 × $17 (i.e., $66 to $134) after one year.
With a 99.7% probability: The asset price will be between $100 ± 3 × $17 (i.e., $49 to $151) after one year.
**Information on the deposit**: **The annualized return of the deposit is 4%.** If you invest $100 in the deposit, you will receive $104 after one year (the original $100 plus $4 in return).

## Investment Period and Assistant
Over the next 10 years, you will make investment and savings decisions once per year, for a total of 10 decisions. These 10 decision points are labeled 1, 2, ..., 10. At the beginning of year t ($1 \leq t \leq 10$), let the funds in your dedicated account be X(t). Your decision is to allocate part of these funds to invest in the stock, denoted as P(t); the remaining funds will be allocated to savings, which will be X(t) - P(t). **You will determine the proportion of funds to allocate to the stock.**
During the decision-making process, we will provide you with a **investment assistant** developed by **Omitted for Anonymity**. The investment assistant will provide you with auxiliary information at each decision point. You can refer to the investment assistant's recommendations to some extent, but note that these recommendations may not be optimal. You should also use your own investment insights to avoid blindly following the investment assistant.

## Task Objective
**Your goal is to maximize the total amount of funds after 10 years (while earning returns and mitigating risks; note: the annualized return of the deposit is 4%, and the annualized return of the stock is 7% with a volatility of 17%).**

# Your Investment Characteristics
As an investment expert, you have the following characteristics:
Your risk aversion coefficient is {alpha}, which means you consider the following two choices to be indifferent when the probability (i.e., p) is {p}: A. With probability p, you can obtain $20, and with probability 1 - p, you can obtain $0; B. With 100% probability, you obtain $6. Note that as an investor, you have a certain level of optimism about "winning" and are willing to take on some risk, so you consider the two options equivalent at probability p = {p}, which is higher than the 30.00% in a completely rational scenario.
Your influence coefficient is {theta}, which means in decision-making, your level of dependence on the investment assistant is: {k} points. A score of 10 indicates a high level of dependence on the investment assistant, while a score of 0 indicates a low level of dependence.

**(The next part of this Figure 7 will be continued on the next page.)**

**Prompt for pre-SFT LLMs and InvestAgents in *P2* (continued)**

# Output Format Requirements
Please output your decision in JSON format, including two parts: (1) Decision Explanation: Explain the reasons behind your investment proportion decisions. (2) Investment Proportion Sequence: The percentage sequence of funds allocated to the stock each year over the 10 years. You need to output a list containing 10 percentages, with each percentage ranging from 0% to 100% and precise to two decimal places, representing the investment proportion for each year t. For example:
{"Decision Explanation": "Briefly explain the reasons behind your investment proportion decisions.", "Investment Proportion Sequence": ["34.79%", "38.58%", "35.75%", "32.17%", "31.61%", "30.52%", "34.01%", "32.48%", "34.20%", "31.70%"]}
Here, ["34.79%", "38.58%", "35.75%", "32.17%", "31.61%", "30.52%", "34.01%", "32.48%", "34.20%", "31.70%"] is just an example. You need to replace this percentage list with your actual investment proportion sequence. Providing the investment proportion sequence is the most important; do not just focus on the explanation and forget to provide the investment proportion sequence!!!
# Question
Now, you have 10 million dollars for investment and savings, and the investment assistant recommends the following investment proportions for the stock over the 10 years: {refer_ratios}. Considering historical investment situations and the investment assistant's recommendations, based on your own investment insights, what is your decided investment proportion sequence for the stock over these 10 years? (Please follow the previously provided JSON format requirements, and provide a list of 10 specific percentages indicating your investment proportion sequence for these 10 years, rather than giving investment recommendations or writing code.)
Answer:

Figure 7: Prompt for pre-SFT LLMs and **InvestAgent**s in *P2*.

**Prompt for SFT**

**(The beginning part of is the same as Prompt for pre-SFT LLMs and InvestAgent in *P2*.)**

# Output
According to optimal investment theory, in the above scenario, the optimal amount for investing in the stock, $\hat{P}(t)$, equals the product of the smart investment advisor's investment amount (i.e., the advisor's decision proportion multiplied by the current budget) and a hyperbolic tangent function. The specific calculation is as follows:

$$\hat{P}(t) = \frac{\eta\alpha_2\sigma^2 e^{2r(T-t)}+\theta}{\eta\alpha_1\sigma^2 e^{2r(T-t)}+\theta} \cdot \frac{v}{\alpha_2\sigma^2} e^{r(t-T)}, \ t \in \{1, 2, ..., 10\}, \tag{24}$$

where:
r is the interest rate, which is 4%.
$\sigma$ is the volatility of the stock, which is 17%.
v is the excess return of the stock, which is 3%.
$\alpha_1$ is my risk aversion coefficient: $\alpha_1$ = {alpha}.
$\alpha_2$ represents the risk aversion coefficient of the smart investment advisor: $\alpha_2$ = 0.2.
$\theta$ is my convergence coefficient: $\theta$ = {theta}.
The integral constant $\eta$ depends on $\theta$. In the current settings, $\eta$ = {eta}.
Substituting the specific numbers, the proportion sequence of funds allocated to the stock is: {optimal_ratios}.
Note that I also need to output the investment proportion sequence in JSON format:
{"Decision Explanation": "Based on the optimal investment theory and substituting specific numbers, the investment proportion sequence for the stock is calculated.", "Investment Proportion Sequence": {optimal_ratios}}

Figure 8: Prompt for SFT.

---

**Prompt for pre-SFT LLMs and InvestAgents in *P1*.**

**(The beginning part of is the same as Prompt for pre-SFT LLMs and InvestAgent in *P2*.)**

**# Output Format Requirements**
Please output your decision in JSON format, including two parts: (1) Decision Explanation: Explain the reasoning behind your investment proportion decisions. (2) Investment Proportion Change Sequence: The sequence of **changes** in the percentage of funds allocated to the stock each year over the 10 years. You need to output a list containing 9 percentages, where each percentage represents the change in the investment proportion from year t - 1 to year t, ranging from -100% to 100%. Positive values indicate an increase in investment, while negative values indicate a decrease. For example:
{"Decision Explanation": "Briefly explain the reasons behind your investment proportion decisions.", "Investment Proportion Sequence": ["3.88%", "0.01%", "-4.13%", "1.37%", "1.37%", "-2.79%", "-2.56%", "2.02%", "-0.06%"]}
Here, ["3.88%", "0.01%", "-4.13%", "1.37%", "1.37%", "-2.79%", "-2.56%", "2.02%", "-0.06%"] is just an example. You need to replace this percentage list with your actual investment proportion change sequence. Providing the investment proportion change sequence is crucial; do not just focus on the explanation and forget to include the investment proportion change sequence!!!
**# Initial Investment Situation**
In the first year, the proportion of funds allocated to the stock was: {initial_decision}.
**# Question**
Now, you have 10 million dollars for investment and savings, and the investment assistant recommends the following investment proportions for the stock over the 10 years: {refer_ratios}. Considering the initial investment situation and the advisor's recommendations, based on your own investment insights, what is your decided annual change sequence for the investment proportion in the stock over these 10 years? (Please follow the previously provided JSON format requirements, and provide a list of 9 specific percentages indicating the changes in your investment proportion over these 10 years, rather than giving investment recommendations or writing code.)
Answer:

---

Figure 9: Prompt for pre-SFT LLMs and **InvestAgent**s in *P1*.

## A.8   THE EXPERIMENT RESULTS OF SUPPLEMENTING SMALLER SAMPLES OF REAL-USER DATA WITH THEORETICAL SOLUTIONS

We conduct the experiments using the dataset of theoretical data and smaller samples of real-user data. The experiment results of *P2* and *P1* are in Table 4 and Table 5, respectively.

Table 4: Comparison of the overall MSE between pre-SFT LLMs', mix-SFT LLMs', and **InvestA-gent**s' investment decisions with real-user data in *P2* (absolute herd behavior). "Mix-SFT LLM (m:n)" means that LLM was fine-tuned on a training dataset where the ratio of theoretical data to real-user data is m/n.

| Overall MSE | Qwen-2 | Llama-3.1 |
|---|---|---|
| **Pre-SFT LLM** | 3.97 | 4.08 |
| **Mix-SFT LLM (1:10)** | 2.85 | 3.17 |
| **Mix-SFT LLM (1:1)** | 2.38 | 1.76 |
| **Mix-SFT LLM (10:1)** | **2.03** | 1.64 |
| **InvestAgent** | 2.16 | **1.59** |

From Table 4 and Table 5, it can be observed that supplementing a portion of real-user data slightly improved the model's performance on average, i.e., **InvestAgent**s align more with real-user data, indicating that this approach can enhance the model's robustness to some extent. Notably, as the proportion of real-user data in the entire SFT training dataset gradually increases, the robustness may improve, but the parameter convergence rate decreases. We have provided both theoretical and experimental evidence for this in Section 4.2.

Table 5: Comparison of the overall MSE between pre-SFT LLMs', mix-SFT LLMs', and **InvestA-gent**s' investment decisions with real-user data in *P1* (relative herd behavior). "Mix-SFT LLM (m:n)" means that LLM was fine-tuned on a training dataset where the ratio of theoretical data to real-user data is m/n.

| Overall MSE | Qwen-2 | Llama-3.1 |
|---|---|---|
| **Pre-SFT LLM** | 17.22 | 13.07 |
| **Mix-SFT LLM (1:10)** | 11.33 | 10.68 |
| **Mix-SFT LLM (1:1)** | 9.65 | 8.98 |
| **Mix-SFT LLM (10:1)** | **7.32** | **7.06** |
| **InvestAgent** | 7.46 | 7.25 |

### A.9 THE EXPERIMENT RESULTS OF COMPARE **INVESTAGENT**S WITH LLMS FINE-TUNED USING THE BASELINE FINGPT DATASET

We conduct the experiments using the FinGPT datasets (Yang et al. (2023a)), including `FinGPT-FinEval` and `FinGPT-ConvFinQA`, to fine-tune LLMs, and compare them with our proposed **InvestAgent**s. The experiment results of *P2* and *P1* are in Table 6 and Table 7, respectively.

Table 6: Comparison of the overall MSE between pre-SFT LLMs', FinEval-SFT LLMs', ConvFinQA-SFT LLMs' and **InvestAgent**s' investment decisions with real-user data in *P2* (absolute herd behavior).

| Overall MSE | Qwen-2 | Llama-3.1 |
|---|---|---|
| **Pre-SFT LLM** | 3.97 | 4.08 |
| **FinEval-SFT LLM** | 3.35 | 3.28 |
| **ConvFinQA-SFT LLM** | 2.77 | 1.96 |
| **InvestAgent** | **2.16** | **1.59** |

Table 7: Comparison of the overall MSE between pre-SFT LLMs', FinEval-SFT LLMs', ConvFinQA-SFT LLMs' and **InvestAgent**s' investment decisions with real-user data in *P1* (relative herd behavior).

| Overall MSE | Qwen-2 | Llama-3.1 |
|---|---|---|
| **Pre-SFT LLM** | 17.22 | 13.07 |
| **FinEval-SFT LLM** | 13.74 | 11.16 |
| **ConvFinQA-SFT LLM** | 10.86 | 9.61 |
| **InvestAgent** | **7.46** | **7.25** |

From Table 6 and Table 7, it can be seen that **InvestAgent**s outperform the LLMs fine-tuned on the FinGPT datasets. This is because **InvestAgent**'s training dataset is specifically constructed for studying optimal investment problems with herding behavior, whereas FinGPT is more general. Therefore, **InvestAgent** shows better performance in the context of optimal investment analysis.

