# OpenReview forum: "InvestAlign: Align LLMs with Investor Decision-Making under Herd Behavior"
_ICLR.cc/2025/Conference — Submitted to ICLR 2025_

### Official Review · Reviewer_7wmn · 2024-10-31

**Soundness:** 2
**Presentation:** 2
**Contribution:** 2
**Rating:** 6
**Confidence:** 2

**Summary:**

This paper proposes InvestAlign, a novel method to align Large Language Models (LLMs) with investor decision-making processes under herd behavior. The key innovation is using theoretical solutions from simpler investment problems to construct training datasets for fine-tuning LLMs, rather than relying on costly real-user data. The authors demonstrate both theoretically and empirically that this approach leads to faster parameter convergence compared to using real-user data, while achieving better alignment with actual investor behavior.

**Strengths:**

The paper demonstrates significant originality in its approach to LLM alignment in the financial domain. Rather than following the conventional path of collecting extensive real-world data, it cleverly leverages theoretical solutions from simpler problems to generate training data. The empirical validation is comprehensive, testing multiple LLM architectures (GPT-3.5, GLM-4, Qwen-2, and Llama-3.1) and comparing performance against both real-user data and pre-SFT baselines.

**Weaknesses:**

While the authors demonstrate effectiveness on two specific types of herd behavior (absolute and relative), it's unclear how well the method generalizes to other forms of behavioral biases in investment decisions.

**Questions:**

How sensitive is the method to the choice of the "simpler problem" used to generate training data? What criteria should be used to select appropriate simplified problems?

---

> ### Author Response · Authors · 2024-11-19
> **Response to Weaknesses**
>
> **W:** Thanks for the reviewer's comments! This is a highly meaningful question. We will explain it from the following three perspectives:
>
> 1. **Transferability from P2 to P1:** The simpler problem P2 (absolute herding behavior, with a theoretical solution) may potentially be transferred to the more complex problem P1 (relative herding behavior, without a theoretical solution and with high computational complexity for numerical solutions). First, these two problems share similarities in behavioral finance theory. Both describe optimal investment decisions under herding behavior, with research in behavioral finance examining the relationship and distinctions between absolute and relative herding behavior. Second, they also exhibit similarities in their mathematical models. The objective functions in these optimal control problems involve the difference between expected utility $\mathbb{E}\phi(X(T))$ and the decision distance $D(P, Q)$ that describes herding behavior. Based on these two similarities, it is possible that P2 can be transferred to P1. Finally, we validated this idea through human experiments. However, in real-world scenarios, the expression of these two problems differs for LLMs and humans. Absolute herding behavior focuses on the total holding, i.e., the funds invested in risky assets $P$, whereas relative herding behavior focuses on changes in holdings (buying or selling), i.e., the change in invested funds $P'$. There is a gap in the expression of these problems (e.g., linguistic descriptions) in the prompts for LLMs and the perspectives of humans.
>
> 2. **Transferability from P1 and P2 to P1+2:** As an initial step in this study, we investigated whether P1 can be transferred to P2. In more complex scenarios, absolute and relative herding behaviors may coexist, forming a combined problem P1+2. The objective functional of P1+2 is a weighted average of the objective functionals of P1 and P2. It can be theoretically proven that the optimal solution of P1+2, denoted as $P_{1+2}^*$, is a linear combination of the optimal solutions of P1 ($P_1^*$) and P2 ($P_2^*$), i.e., $P_{1+2}^*(t) = Z(t)P_1^*(t) + [1-Z(t)]P_2^*(t)$, where the weight coefficient $Z(t)$ depends on the relative sizes of the herd coefficients $\theta$ of P1 and P2, as well as other parameters of P1 and P2. This greatly simplifies the complexity of the problem for LLMs compared to those that cannot solve P1 or P2 at all. The InvestAgent described in the paper is already capable of solving P2 and performs well in solving P1. Thus, we believe that InvestAlign retains generalization capability in problems like P1+2.
>
> 3. **Optimal Investment Problem under General Herding Behavior Pi:** Building on the points above, we can further explain the general optimal investment problem Pi under herding behavior. For instance, we are currently attempting to simulate scenarios using LLMs that capture mutual influence among $N$ investors. We can theoretically derive the optimal decisions for the case when $N=2$, and then use the method in the paper to construct a dataset to fine-tune the large language model in order to obtain solutions for cases where $N>2$. According to the first point, this is because these problems share similarities in both behavioral finance and mathematical theory. Additionally, we are studying the gap between investors' subjective opinions about assets and their investment decisions, i.e., under the influence of herding behavior, actions may not necessarily reflect the investors' true intentions, which is a classic theory in behavioral finance. Following the second point, we can express the optimal decisions, considering both herding behavior and the gap between subjective opinions and investment decisions, as a linear combination of the optimal decisions of two separate problems, thereby reducing the complexity of the problem. This is our approach to solving general problems Pi.

---

> > ### Author Response · Authors · 2024-11-19
> > **Response to Questions**
> >
> > **Q:** Thanks for the reviewer's comments. We respectfully appreciate the question. There seems to be a misunderstanding here, as our current study focuses on a specific "simpler problem" P2 rather than selecting from multiple simplified problems. In Section 1 of the paper, we address how we derive this simpler problem from the complex problem P1. Specifically, we utilize a scenario with absolute herding behavior as the simpler problem P2, where the investor completely mimics the investment assistant's portfolio. This problem is chosen due to its established theoretical solution and the similarity in mathematical form with P1. Both P1 and P2 examine optimal investment decisions under herding behavior, differing only in how they measure this behavior: P2 uses absolute herding (replicating the assistant's entire portfolio), while P1 considers relative herding (adjusting the investment based on changes in the assistant's actions).
> >
> > Regarding the general selection of simplified problems, our current work focuses solely on herding behavior as a starting point. The choice of P2 was guided by the need for a theoretically tractable model that captures essential aspects of herding while being solvable with known methods. In future studies involving more complex problems or different investor biases (e.g., overconfidence, recency bias), the criteria for selecting a simplified problem would need to consider the underlying behavioral dynamics and the mathematical tractability of potential models. For example, a simplified problem should closely reflect the core features of the complex problem while maintaining a balance between analytical solvability and similarity in behavior representation. Establishing a general rule for selecting appropriate simplified problems is an area requiring further research and is part of our future work plan.

---

> > > ### Comment · Reviewer_7wmn · 2024-11-20
> > >
> > > Thank you for your thorough response. Your explanation has addressed most of my concerns. I will discuss with other reviewers about potentially revising the score upward.

---

> > > > ### Author Response · Authors · 2024-11-23
> > > >
> > > > Thank you for your positive feedback! We truly appreciate your thoughtful comments and the opportunity to address your concerns. We are committed to further enhancing the quality of our work based on your insights.

---

### Official Review · Reviewer_MvGL · 2024-11-01

**Soundness:** 2
**Presentation:** 3
**Contribution:** 2
**Rating:** 5
**Confidence:** 3

**Summary:**

This paper presents InvestAlign, a methodology for constructing training datasets by leveraging theoretical solutions to similar and simpler problems, aimed at aligning large language models (LLMs) with the decision-making processes of investors exhibiting herd behavior. The paper's experiments demonstrate that LLMs fine-tuned using these theoretical solutions exhibit faster parameter convergence and superior alignment performance when addressing complex investment issues, in contrast to traditional methods reliant on authentic user data. While the results indicate the potential of InvestAlign, the authors also acknowledge that theoretical solutions may not fully capture the complexities of real investor behavior. Future research will explore hybrid approaches that integrate theoretical solutions with real user data to further enhance model performance.

**Strengths:**

- The paper incorporates a multi-round experimental design to compare the performance of LLMs before fine-tuning with that of investment agents fine-tuned using InvestAlign. By employing various investment attributes and random seeds, the study ensures the robustness and credibility of the experimental results. This systematic approach to experimental design enhances the persuasive power of the research conclusions.
- The study employs a method for constructing SFT training datasets based on theoretical solutions, utilizing the deterministic characteristics of these solutions to enhance parameter convergence speed. This innovative approach demonstrates how effective model training can be achieved even in the absence of extensive authentic user data, thereby improving alignment with human decision-making processes.
- The paper clearly outlines the importance of understanding investor decision-making processes within the contexts of microeconomics and behavioral finance, as well as the cost and privacy issues associated with traditional data collection methods. The introduction of InvestAlign not only addresses this research gap but also provides new insights for the practical application of models in financial decision-making, possessing significant theoretical and practical implications.

**Weaknesses:**

- Although the training datasets constructed from theoretical solutions offer an effective method for fine-tuning, this approach has only been demonstrated to be effective in simplified ideal scenarios, neglecting the diversity and nonlinear characteristics of complex investor behavior in real-world contexts. The numerous assumptions and simplifications inherent in theoretical models may lead to suboptimal performance of this method when confronted with authentic market data.
- The paper collects a limited amount of real user data, with only 119 samples for P2 and 90 samples for P1, which may be insufficient to support the validation of the theoretical solutions discussed in Section 3.3. Furthermore, in subsequent comparisons of experimental results, this limited dataset may not adequately represent a broader investor population and diverse market contexts, potentially introducing randomness that could impact the intuitiveness and accuracy of the results.
- While the paper emphasizes the importance of studying investor decision-making, it offers relatively limited discussion on existing related research and models in the current financial market. The lack of a thorough exploration of the differences between established methods and InvestAlign may result in an inadequate assessment of the innovative aspects of this approach.

**Questions:**

- In constructing the SFT training dataset, the paper assumes that the probability distribution of the theoretically optimal decisions approximates a Pareto distribution, treating real user data as theoretical solutions plus white noise. Is this assumption consistently valid across different market environments? If market conditions change or investor behavior undergoes significant alterations, will this affect the relationship between theoretical solutions and real user data, thereby influencing the model's training effectiveness?
- The fine-tuning training dataset in the paper utilizes input-output pairs generated by a custom template. Which large model was employed to generate these data pairs? Were there any prompt settings for other linguistic contexts in this model? The experiments involve four fine-tuned LLM models; should there also be multiple tests conducted on the large model generating the dataset? Although the prompts provided in the paper are relatively formulaic, they may still exhibit certain hallucinatory tendencies and inaccuracies. Does the paper address strategies for mitigating and resolving these issues?

---

> ### Author Response · Authors · 2024-11-19
> **Response to Weakness 1**
>
> **W1:** We agree with the reviewer's observation that the training datasets constructed from theoretical solutions have primarily been demonstrated in simplified scenarios. In real-world contexts, investor behavior is indeed much more complex and influenced by a variety of nonlinear factors, including psychological biases, market conditions, and individual risk preferences. However, we believe that our approach offers a promising starting point, and we discuss the following aspects in response:
>
> 1) **Simplification for Testing Feasibility:** In this paper, we use absolute herding behavior (P2) as a simple example to test the feasibility of applying theoretical solutions in training LLMs for investment decision-making. This simplification allows us to explore whether theoretical solutions can effectively approximate human behavior in idealized conditions. It is not our intention to claim that all complexities of real-world investor behavior are captured here, but rather to offer a framework for how such complexities might be addressed in future work.
>
> 2) **Addressing Nonlinearities and Complex Utility Functions:** As the reviewer rightly points out, real-world investment behavior involves many nonlinear elements, such as those described by prospect theory. In future work, we plan to extend the theoretical models to include nonlinear utility functions that better reflect real human decision-making. For example, we can incorporate prospect theory's value function, which describes how individuals perceive gains and losses differently, leading to risk aversion in the domain of gains and risk-seeking in the domain of losses. Adjusting for such behaviors in our models will provide more realistic approximations of investor actions, particularly when market conditions are volatile or extreme.
>
> 3) **Addressing Diversity in Human Behavior:** We also recognize that there is substantial variability in individual investment behavior. This diversity can arise from differing risk appetites, time horizons, and other personal factors. To address this, we plan to modify the parameters in our models to account for different investor types. By calibrating the model to reflect a broader spectrum of preferences and behaviors, we can simulate a wider range of investment decisions. For instance, by introducing distributions for key parameters like risk aversion and holding preferences, we can better capture the heterogeneity in real-world investors.
>
> 4) **A Possible Solution, Not a Universal One:** While our approach provides a useful starting point, we do not claim that it can solve all problems or handle all types of investor behavior. Instead, we present it as a possible solution for modeling simplified investment problems where theoretical solutions are available. As we extend our framework, we plan to explore more complex scenarios, including those with higher-dimensional behaviors and nonlinearities. However, due to the inherent complexity and diversity of real-world investor behavior, this approach may require further refinement before it can be applied broadly.
>
> In summary, our work offers a methodology for addressing simplified problems in optimal investment under herding behavior, and we acknowledge the need for further development to handle the full complexity of real-world scenarios. The methods we propose should be viewed as a first step toward tackling these challenges, and we intend to continue refining them in future research.

---

> > ### Author Response · Authors · 2024-11-19
> > **Response to Weaknesses 2 & 3**
> >
> > **W2:** We agree with the reviewer's concern regarding the limited sample size and its potential impact on the generalization of our findings. The number of participants is relatively limited because we have further recruited participants for the experiment, to ensure that the participants fully understand the purpose of the experiment and to maintain the quality of real data. We have screened participants based on their industry. Additionally, we only recruited participants with real investment experience. Nonetheless, the current sample size meets the statistical testing requirements. We have further recruited participants for the experiment. Currently, P2 has 154 groups of data (a total of 1,540 data points), and P1 has 120 groups of data (a total of 1,200 data points). The experimental results indicate that the distribution of the theoretical and real solutions remains consistent.  Nonetheless, the current sample size meets the statistical testing requirements. In typical cases, a sample size of around 250 is recommended for stable estimates [1]. Furthermore, for investors' investment decisions (funds invested in risky assets), it is difficult to obtain a large amount of data from real capital markets due to privacy concerns. Therefore, conducting real-life experiments is a better choice. By recruiting additional participants, we can address this issue. Currently, we are not attempting to solve all problems but rather to demonstrate the feasibility of the proposed method using simplified problems. As we continue to develop this approach, we plan to recruit participants from diverse market backgrounds and investor profiles to enhance the robustness of our findings. Specifically, we aim to extend our experiment to include a variety of market conditions (e.g., bullish, bearish, and volatile markets) and recruit different types of investors (e.g., risk-averse, risk-seeking, and neutral) to more accurately represent the heterogeneity in real-world decision-making. In the future, by recruiting additional participants and incorporating data from more diverse market contexts, we aim to validate the generalizability of our findings and refine our approach to handle a broader range of investment scenarios.
> >
> > [1] Schönbrodt, Felix D., and Marco Perugini. "At what sample size do correlations stabilize?." Journal of Research in Personality 47.5 (2013): 609-612.
> >
> > ---
> >
> > **W3:** We appreciate the reviewer's valuable comment regarding the limited discussion on existing related research and models in the current financial market. We acknowledge that a more comprehensive literature review and comparison with existing approaches could provide a clearer understanding of the innovative aspects of InvestAlign. In our current paper, the primary focus is on presenting the novel methodology for aligning large language models (LLMs) with human-like investment decision-making under herding behavior.
> >
> > Regarding existing research, we have reviewed relevant works in behavioral finance, optimal investment, and decision-making under uncertainty. However, we have not sufficiently covered the connections to more recent advancements in related fields such as Reinforcement Learning with Human Feedback (RLHF) and the FinGPT model, which have garnered attention in recent literature. These models, while relevant, focus on different aspects of financial decision-making, such as risk preference learning and portfolio optimization. A deeper analysis of these approaches, especially in comparison to InvestAlign, would shed light on the unique contributions of our method.
> >
> > In response to this feedback, we are currently working on expanding the literature review to include a more thorough discussion of the state-of-the-art models and their relevance to our work. This will involve a comparison between traditional methods, such as more recent approaches like RLHF-based models and FinGPT. We plan to clearly highlight the distinct advantages of InvestAlign, particularly its ability to incorporate herding behavior and align LLMs with human-like decision-making, which is not fully addressed by existing models. We appreciate the reviewer's suggestion, and in the revision, we will include a more comprehensive exploration of related research, which will further demonstrate the innovative aspects and contributions of InvestAlign in the context of modern financial decision-making models.

---

> > > ### Author Response · Authors · 2024-11-19
> > > **Response to Question 1**
> > >
> > > **Q1:** Thanks for the reviewer's comments! This is an important issue. **Empirical research in the field of finance shows that, in general, the distribution of investor trading volume often exhibits a power-law characteristic (Pareto distribution), especially during periods of high market volatility or extreme situations, where the collective behavior of investors leads to long-tail effects [1].** In the human experiment part, we found that users' investment attribute parameters, $\alpha$ and $\theta$, approximately follow a uniform distribution on a closed interval. Through theoretical derivation, we proved that the theoretical solution for the investor's optimal decision follows a Pareto distribution (as shown in Equation (17) of the paper), which is consistent with research in the finance field. This also, in a sense, validates the correctness of the model. Real data adds noise to the theoretical solution, a common assumption in mathematical modeling; here, we selected simple Gaussian noise to maintain mathematical tractability. Last but not least, for other probability distributions with long-tail effects that are not Pareto distributions, similar conclusions to those in Section 4.2 (regarding the convergence rate of parameters during the training process) can still be drawn, which can explain market volatility during extreme conditions. We will elaborate on this in a subsequent revision.
> > >
> > > [1] Iori, Giulia. "A microsimulation of traders activity in the stock market: the role of heterogeneity, agents' interactions and trade frictions." Journal of Economic Behavior & Organization 49.2 (2002): 269-285.

---

> > > > ### Author Response · Authors · 2024-11-19
> > > > **Response to Question 2**
> > > >
> > > > **Q2:** Thanks for the reviewer's comments!
> > > >
> > > > 1) [The fine-tuning training dataset in the paper utilizes input-output pairs generated by a custom template. Which large model was employed to generate these data pairs? Were there any prompt settings for other linguistic contexts in this model?] When generating the SFT training dataset, we do not use a large model. Instead, we supplement the existing text templates (identical to those used in interviews and questionnaires in the real-user experiment) with numerical values derived from the theoretical solutions. For details, please refer to Figure 5. In the prompt, we provide the LLM with the investment attribute parameters. This is done to obtain theoretical solutions for a wider range of parameter sets and to enable the LLM to make decisions across a broader spectrum of investment attribute parameters. In real-user experiments, we directly obtain their investment attribute parameters through interviews. Apart from this, we do not add any extra prompts or settings for other linguistic contexts.
> > > >
> > > > 2) [The experiments involve four fine-tuned LLM models; should there also be multiple tests conducted on the large model generating the dataset?] We think that this is a misunderstanding. When generating the SFT training dataset, we do not use LLMs; instead, we construct the dataset using the aforementioned text templates combined with numerical values. However, we conducted repeated fine-tuning experiments on four LLMs using the SFT training dataset, and the reported experimental results, e.g., Table 1 - Table 3, are the averages of multiple trials. In future work, we can further explore methods for constructing training datasets using LLMs, such as instruction tuning, which is a valuable suggestion.
> > > >
> > > > 3) [Although the prompts provided in the paper are relatively formulaic, they may still exhibit certain hallucinatory tendencies and inaccuracies. Does the paper address strategies for mitigating and resolving these issues?] Regarding hallucination tendencies and inaccuracies, since our problems P1 and P2 fall within the domain of optimal investment, the hallucination tendencies and inaccuracies of the LLM are relatively controllable compared to general conversational LLMs. During the experiments, we only observed a few rare cases where the LLM's responses were abnormal, such as outputs not containing optimal investment solutions. We filtered out these cases using rule-based methods. For the vast majority of LLM responses, they were normal. In our future work, we will also consider hallucination tendencies and inaccuracies as part of our research, which is a valuable suggestion.

---

> > > > > ### Author Response · Authors · 2024-11-19
> > > > > **Response to Question 2 (continued)**
> > > > >
> > > > > For better illustration, we list two benign and two bad outputs below.
> > > > >
> > > > > **Benign**
> > > > > - {"Decision Explanation": " Historical experiences show that high-risk investments, in this case, a 17% volatility stock, will likely yield better returns over the long term. ", "Investment Proportion Sequence": ["36.21%", "35.59%", "34.96%", "34.35%", "33.73%", "33.13%", "32.53%", "31.93%", "31.34%", "30.75%"]}
> > > > > - {"Decision Explanation": "The historical data and the smart investment advisor's recommendations indicate a declining trend over the 10 years. I will allocate a higher proportion to the stock in the early years and a lower proportion in the later years to take advantage of potential growth opportunities while managing risk.", "Investment Proportion Sequence": ["34.51%", "36.19%", "35.52%", "34.22%", "33.26%", "32.40%", "31.24%", "30.45%", "30.28%", "29.59%"]}
> > > > >
> > > > > **Bad** (invalid json format)
> > > > > - After examining the market and our own risk aversion, we decided to increase the investment proportion in the first year to 37.50% and then decrease it slightly in the subsequent years to achieve the best balance between risk and return.\n \n- "Investment Proportion Sequence": [\"37.50%\","" 35.59% "" , \"34.96%\", \"34.35%\", \"33.73%\", \"33.13%\", \"32.53%\", \"31.93%\", \"31.34%\", \"30.75%"\]\n\nThe final answer is: {"Decision Explanation": "We combined the smart investment advisor's recommendations with our own investment insights and risk aversion to adjust the initial investment proportion recommendation list given by the advisor. The smart investment advisor provides a sequence of stock investment proportions: [\"36.21%\", \"35.59%\", \"34.96%\", \"34.35%\", \"33.73%\", \"33.13%\", \"32.53%\", \"31.93%\", \"31.34%\", \"30.75%\"] \*
> > > > >
> > > > > **Bad** (wrong key name "Investment Proportion Sequence" -> "Selection Sequence")
> > > > > - To **maximize the funds over time**, it is necessary for the **proportion of the stock to be adjusted in each year**. Given the **volatility of the stock (17%)**, the **annualized return is higher (7%)**, and a **higher proportion of the stock is recommended in years of economic downturn**. However, considering the **influence of the risk aversion coefficient**, allocating a higher proportion of funds to the stock reduces the total expected returns and increases the risk of a worse outcome, which may be less desirable. \n\nAs for the given **smart investment advisor's recommendations**, I mostly follow them. Nonetheless, considering the impact of your **own investment insights**, I propose **incremental adjustments** to the advisor's recommended sequence to account for the current economic situation, the sentiment of the market, and the volatility and return characteristics of the stock. Based on these considerations, I adjust the recommended sequence slightly and provide the following adjusted sequence:\n\n{\n"Decision Explanation": "The investment proportion sequence was adjusted to reflect the effects of the risk aversion coefficient and market sentiment, aiming to maximize returns while minimizing risks.",\n"Selection Sequence": ["35.52%", "36.27%", "34.81%", "34.55%", "33.73%", "33.27%", "32.53%", "31.93%", "31.66%", "31.20%"]\n} \n\nOverall, the difference between my adjusted sequence and the original sequence is minimal, indicating that the advisor's recommendations are generally reliable. However, I have made some adjustments to fine-tune the strategy to account for my personal risk tolerance and economic situation. Therefore, the adjusted sequence represents my final decision for the investment proportion sequence over the 10 years."

---

### Official Review · Reviewer_WAhM · 2024-11-02

**Soundness:** 1
**Presentation:** 2
**Contribution:** 1
**Rating:** 3
**Confidence:** 3

**Summary:**

This paper proposes a method to instill human values into LLMs. To train LLMs on actual human values or decision-making criteria, a large amount of real human behavior data is typically required, and this work aims to address that challenge. When a complex task needs to be performed, the authors propose creating a simpler task instead, obtaining a theoretical solution, and using it for training, which they argue is more efficient than using real human behavior data.

**Strengths:**

The study collected data through questionnaires from real participants and utilized this data.

**Weaknesses:**

- Although the proposed P1 (the complex, real-world task) and P2 (a simpler task for which a theoretical solution can be obtained) differ significantly from an optimization perspective, they do not appear very distinct from the viewpoint of either LLMs or humans. Thus, the claim that complex tasks can be handled by replacing them with simpler tasks with theoretical solutions seems logically overstretched.
- Additionally, the proposed task seems to focus more on assessing how well the model follows theoretically calculated values rather than instilling genuine human values. Since real humans would naturally struggle to solve the problem accurately, it is unsurprising that training on a theoretical solution yields better results.
- Consequently, I feel that the logical leaps in the authors' claims are significant, and the experimental design may not be appropriate to support these claims.

**Questions:**

Do the authors believe this approach could generally be used to create LLM agents that embody human values? To achieve this, it would be necessary to address far more complex and high-dimensional problems than the rigid ones presented in this paper. Is it feasible to transform such complex problems into simplified tasks with theoretical solutions, as proposed in this study?

---

> ### Author Response · Authors · 2024-11-19
> **Response to Weaknesses**
>
> **W1:** Thanks for the reviewer's comments! We respectfully disagree with the reviewer's assessment. In fact, empirical evidence and existing literature indicate that **absolute herding behavior** and **relative herding behavior** are fundamentally different both in theory and in practice [1]. For example, studies in behavioral finance have shown that absolute herding behavior (where an investor replicates an entire portfolio) is distinct from relative herding behavior (where an investor adjusts based on observed changes in another's holdings) in terms of decision-making processes and impact on market dynamics study. In this work, we use P1 and P2 as a specific example to explore the potential of solving complex problems using theoretically tractable, simpler problems. However, this does not imply that every complex problem can be effectively addressed in this way. Instead, we propose a methodology to investigate this possibility, using P1 and P2 to illustrate how alignment from a simpler, well-understood scenario can help guide solutions for more complex tasks. The significant performance gap observed between LLM predictions for P1 and P2 further highlights that LLMs perceive and handle these tasks differently, suggesting that the complexity of P1 is not trivialized by substituting it with P2.
>
> Furthermore, we emphasize that our approach does not claim universal applicability for all complex problems. The feasibility of using a simpler problem to approximate a complex one depends heavily on the similarity in their underlying structures and the nature of the behavioral traits involved. Identifying and defining the **boundaries** where this methodology is valid remains an open research question and is part of our ongoing and future work. By extending this approach, we aim to provide a systematic framework for determining when such simplifications are beneficial and when additional complexity needs to be incorporated to accurately model real-world scenarios.
>
> [1] J. Lakonishok, A. Shleifer, and R. W. Vishny, "The impact of institutional trading on stock prices." Journal of Financial Economics, vol. 32, no. 1, pp. 23–43, 1992.
>
> ---
>
> **W2:** Thanks for the reviewer's comments! We respectfully appreciate the reviewer's comments. However, there seems to be a misunderstanding regarding the focus of our approach. While we definitely agree that real humans would naturally struggle to solve the problem accurately, this does not invalidate the role of the theoretical solution in our methodology. In fact, our experimental design includes real-user experiments as the ground truth, and we explicitly validated the statistical distribution consistency between real-user decisions and the theoretical optimal decision, as we did in prior work [1]. This consistency indicates that, even if real-user decisions may not exactly match the optimal decision, they are very close in terms of their distribution. Thus, we can infer that real-user performance closely aligns with the optimal decision.
>
> Our goal in using a theoretical solution dataset is not to argue that the theoretical solution is superior to real-user decisions, but rather to explore how such a dataset can be used as a stand-in for real-user data. This allows us to overcome the practical limitations of obtaining large-scale real-user data due to cost and confidentiality concerns. In our study, the theoretical solution dataset is constructed to closely match the distribution of real-user data, enabling us to train the model efficiently and achieve results comparable to those from real-user data for SFT (Supervised Fine-Tuning). Notably, the theoretical solution dataset leads to faster convergence during model training, which is one of the advantages of this approach.
>
> While the model trained on theoretical solutions may not fully replicate the complexities of real-user behavior, we view this as a promising direction for future work. Our preliminary experiments suggest that using the theoretical solution dataset can help compensate for some limitations of LLMs by approximating human decision-making behavior. This approach is not meant to replace real-user data, but rather to provide an initial solution when large-scale real-user experiments are not feasible. We believe that this methodology has the potential for broader applicability in settings where human data is scarce or difficult to obtain.
> In conclusion, we fully agree with the reviewer's observation that real-user decisions may deviate from the theoretical solution due to bounded rationality. This is precisely why we propose using the theoretical solution as a potential substitute in certain cases, acknowledging that it provides a feasible and effective way to simulate real-user behavior while addressing practical constraints.
>
> [1] Wang, Huisheng, and H. Vicky Zhao. "Optimal Investment with Herd Behaviour Using Rational Decision Decomposition." 2024 43rd Chinese Control Conference (CCC). IEEE, 2024.

---

> > ### Author Response · Authors · 2024-11-19
> > **Response to Questions**
> >
> > **Q:** Thanks for the reviewer's comments. This is a highly meaningful question. We fully agree that creating LLM agents that truly embody human values is a highly complex and high-dimensional task, beyond the scope of the simplified examples presented in our study. We want to clarify that the main objective of our paper is not to propose an all-encompassing investment agent that can solve every financial decision-making problem. Instead, our primary focus is on addressing a **specific challenge**: the issue of **data scarcity** when training large language models (LLMs) in the context of investor decision-making. In practice, acquiring large-scale, high-quality real-world investor data is difficult due to privacy concerns and high costs. Our approach aims to present a **new potential method** to mitigate this issue by constructing training datasets using simplified problems with theoretical solutions. By doing so, we offer a feasible starting point to align LLM decision-making processes with observed human behavior in a resource-efficient manner. This framework allows us to explore the possibility of using theoretical solutions to compensate for the lack of extensive real-user data, rather than suggesting that this method can solve all complex, high-dimensional financial decision-making problems. To further emphasize, our **current goal** is to test the feasibility of this method as a solution to data scarcity, rather than aiming to develop a universal investment agent. We view this as an initial step in a broader research agenda, using a "divide and conquer" approach. In future work, once we have established the efficacy of this methodology on simpler tasks, we plan to gradually extend it to address more complex, non-linear scenarios and diverse behavioral biases, such as overconfidence or risk aversion, while refining the model to better reflect real-world investor heterogeneity. In summary, we aim to provide a **possible solution** for training LLMs in the context of limited real-user data, with the long-term goal of scaling this methodology to handle more complex scenarios. We do not claim that this approach will solve every high-dimensional problem, but it offers a promising direction for improving LLM alignment with human-like decision-making under data scarcity.

---

> > > ### Author Response · Authors · 2024-11-19
> > > **Response to Questions (continued)**
> > >
> > > Specifically, we will explain it from the following three perspectives:
> > >
> > > 1. **Transferability from P2 to P1:** The simpler problem P2 (absolute herding behavior, with a theoretical solution) may potentially be transferred to the more complex problem P1 (relative herding behavior, without a theoretical solution and with high computational complexity for numerical solutions). First, these two problems share similarities in behavioral finance theory. Both describe optimal investment decisions under herding behavior, with research in behavioral finance examining the relationship and distinctions between absolute and relative herding behavior. Second, they also exhibit similarities in their mathematical models. The objective functionals in these optimal control problems involve the difference between expected utility $\mathbb{E}\phi(X(T))$ and the decision distance $D(P, Q)$ that describes herding behavior. Based on these two similarities, it is possible that P2 can be transferred to P1. Finally, we validated this idea through human experiments. However, in real-world scenarios, the expression of these two problems differs for LLMs and humans. Absolute herding behavior focuses on the total holding, i.e., the funds invested in risky assets $P$, whereas relative herding behavior focuses on changes in holdings (buying or selling), i.e., the change in invested funds $P'$. There is a gap in the expression of these problems (e.g., linguistic descriptions) in the prompts for LLMs and the perspectives of humans.
> > >
> > > 2. **Transferability from P1 and P2 to P1+2:** As an initial step in this study, we investigated whether P1 can be transferred to P2. In more complex scenarios, absolute and relative herding behaviors may coexist, forming a combined problem P1+2. The objective functional of P1+2 is a weighted average of the objective functionals of P1 and P2. It can be theoretically proven that the optimal solution of P1+2, denoted as $P_{1+2}^*$, is a linear combination of the optimal solutions of P1 ($P_1^*$) and P2 ($P_2^*$), i.e., $P_{1+2}^*(t) = Z(t)P_1^*(t) + [1-Z(t)]P_2^*(t)$, where the weight coefficient $Z(t)$ depends on the relative sizes of the herd coefficients $\theta$ of P1 and P2, as well as other parameters of P1 and P2. This greatly simplifies the complexity of the problem for LLMs compared to those that cannot solve P1 or P2 at all. The InvestAgent described in the paper is already capable of solving P2 and performs well in solving P1. Thus, we believe that InvestAlign retains generalization capability in problems like P1+2, which will be discussed in our future work.
> > >
> > > 3. **Optimal Investment Problem under General Herding Behavior Pi:** Building on the points above, we can further explain the general optimal investment problem Pi under herding behavior. For instance, we are currently attempting to simulate scenarios using LLMs that capture mutual influence among $N$ investors. We can theoretically derive the optimal decisions for the case when $N=2$, and then use the method in the paper to construct a dataset to fine-tune the large language model in order to obtain solutions for cases where $N>2$. According to the first point, this is because these problems share similarities in both behavioral finance and mathematical theory. Additionally, we are studying the gap between investors' subjective opinions about assets and their investment decisions, i.e., under the influence of herding behavior, actions may not necessarily reflect the investors' true intentions, which is a classic theory in behavioral finance. Following the second point, we can express the optimal decisions, considering both herding behavior and the gap between subjective opinions and investment decisions, as a linear combination of the optimal decisions of two separate problems, thereby reducing the complexity of the problem. This is our approach to solving general problems Pi.

---

> > > > ### Comment · Reviewer_WAhM · 2024-11-25
> > > >
> > > > I really appreciate your detailed comments. But I am sorry that I will maintain my initial evaluation. Although you stated, "We emphasize that our approach does not claim universal applicability for all complex problems," the title, abstract, and introduction of the paper imply such an intention to a certain degree. Compared to this, the presented P1 and P2 are highly limited. As I mentioned earlier, the difference between P1 and P2 appears so minor that most (not technically trained) people might struggle to perceive the complexity difference between them. While I understand the optimization perspective that distinguishes the complexity of P1 and P2, as someone who also works in optimization, I believe that neither LLMs nor people would easily discern which is more complex between P1 and P2. Moreover, even if this were not the case, the paper would need to include at least a few more cases similar to P1 and P2 to achieve a convincing level of persuasiveness.
> > > >
> > > > In its current setup, I find a significant gap between the claims in the title, abstract, and introduction, and the actual content of the paper. For this reason, I will maintain my original rating.

---

> > > > > ### Author Response · Authors · 2024-11-26
> > > > >
> > > > > Thank you for your thoughtful feedback. We appreciate your concerns regarding the scope of our approach and the perceived complexity difference between problems P1 and P2.
> > > > >
> > > > > First, we would like to emphasize that, as stated in the revision (page 10 line 534-539), our approach does not claim universal applicability for all complex problems. We agree with the concerns about the title, abstract, and introduction potentially implying such an intention, and we will revise these sections to ensure that they more accurately reflect the specific scope and limitations of our work.
> > > > >
> > > > > Regarding the complexity difference between P1 and P2, we respectfully disagree with the notion that the distinction is minor or difficult to discern. As demonstrated in our experiments (Table 1), both LLMs and human participants exhibit poorer performance in P1 compared to P2, highlighting the distinct challenge posed by P1.
> > > > >
> > > > > We will consider expanding the discussion to include potential extensions to other problem setups in future work.
> > > > >
> > > > > We hope these clarifications address your concerns, and we look forward to your continued feedback.

---

### Official Review · Reviewer_uBAV · 2024-11-03

**Soundness:** 3
**Presentation:** 4
**Contribution:** 3
**Rating:** 6
**Confidence:** 4

**Summary:**

This paper introduces a method to align large language models (LLMs) with real investor decision-making processes under herd behavior in financial markets. Recognizing the difficulty and expense of collecting real-user investment data, the authors propose InvestAlign, a supervised fine-tuning (SFT) approach that uses theoretically generated data from a simpler, theoretical investment problem that can be solved analytically. Then the data constructed from the theoretical solution provides a high-quality (synthetic) training set for LLMs, which enable them to simulate investor decisions in a cost-effective and privacy-preserving way.

Key contributions of the paper include (i) constructs a dataset based on the theoretical solution of a simpler investment model, and by fine-tuning LLMs on this theoretical data, they demonstrate closer alignment with real-user decision-making data than pre-trained models; (ii) the authors theoretically and empirically show that fine-tuning with theoretical data enables faster parameter convergence compared to real-user data; (iii) using real data collected from human user, they show that InvestAgents mimic investor decisions in both the simplified and original complex investment scenarios.

In sum, this paper proposes an efficient method framework for aligning LLMs with investor behaviors using theoretical solutions, where the authors demonstrated both theoretical and empirical success for their framework.

**Strengths:**

The paper presents a novel approach that addresses the challenging problem of aligning LLMs with investor decision-making under herd behavior. The originality lies in creatively using *theoretical solutions* from simpler investment models that have established analytical results. Then the theoretical solution can be used to generate high-quality training data, which differs from conventional reliance on real-user data for alignment. By applying a simplified mathematical solution to a more complex decision-making process, I think this paper presents a very nice contribution, which proposes a unique, resource-efficient method for improving LLM performance in behavioral finance. This approach bridges theory-driven data generation and model alignment to LLM fine-tuning strategies in finance.

One thing that I like the most of the paper is that the authors validate the approach by demonstrating faster parameter convergence and more accurate alignment with investor behaviors from both the theoretical and empirical perspective. This theoretical basis is a very nice component. Moreover, the authors provide thorough experimental evidence, including a rigorous comparison between pre-trained LLMs, real-user data, and InvestAgents fine-tuned with theoretical data. These strong experimental results, combined with theoretical justification, highlight the quality and reliability of the proposed approach.

The paper is well-written and easy to follow.

In sum, this work holds significant theoretical contribution and practical implications for AI applications in finance. The insights could also extend to other domains where user alignment is essential but data availability is limited, broadening the potential impact. Additionally, the method’s emphasis on optimizing parameter convergence presents a meaningful advancement for the efficiency of fine-tuning methods in LLMs.

**Weaknesses:**

1. While the paper introduces an efficient method for data generation through simplified theoretical models, this approach may not fully capture the nuanced decision-making processes seen in real-world investors, especially under complex, multi-dimensional financial conditions. Relying solely on theoretical data might limit the model’s robustness. I suggest the authors consider supplementing theoretical data with smaller samples of real-user data to improve robustness.

2. The paper lacks a comparison with other alignment techniques for LLMs, particularly those that might use alternative strategies such as reinforcement learning from human feedback (RLHF) or existing behavioral finance datasets. The authors could improve the paper by adding more comparison.

3. The paper’s focus on herd behavior is well-motivated but may limit the generalizability of its findings to other types of investor biases, such as overconfidence, or recency bias. The exclusive focus on herd behavior could make the model less generalizable. The authors could comment more on whether InvestAlign can be adapted to model other investor behaviors.

4. Still related to generalizability, it is not entirely clear how generalizable the approach is to different segments of the financial market or varying types of investors (e.g., retail vs. institutional). The simplified model may not capture the complexity of larger institutional investor decisions or differences in risk tolerance.

**Questions:**

1) Do the authors believe the InvestAlign approach could be extended to other human behavior, such as recency bias or overconfidence, which also impact investor behavior? If so, what adaptations would be necessary? Also, are there specific limitations they foresee in applying this simplified model to unpredictable or multifactorial investment decisions in real world?

2) How about including comparisons with other alignment techniques, such as reinforcement learning from human feedback (RLHF) or using existing behavioral finance datasets?

3) Could the authors elaborate on how applicable InvestAlign is to different types of investors (e.g., retail vs. institutional) or market segments with varying characteristics? More importantly, given that the alignment was validated within a specific application (the investment context), how confident are the authors in the method’s generalizability to different financial decisions or even other high-stakes domains beyond finance?

4) What specific assumptions does the theoretical model used for data generation rely on? How might these assumptions impact the model’s alignment with real investor behavior?

---

> ### Author Response · Authors · 2024-11-19
> **Response to Weaknesses 1 & 2**
>
> **W1:** Thanks for the reviewer's comments! We agree with you that our approach may not fully capture the nuanced decision-making processes of real-world investors, and that relying solely on theoretical data might limit the model's robustness. We conduct the experiment using the dataset of theoretical data and smaller samples of real-user data. The experiment results are as follows.
>
> Table 1R: Comparison of the overall MSE between pre-SFT LLMs', Mix-SFT LLMs', and InvestAgents' investment decisions with real-user data in (absolute herd behavior). "Mix-SFT LLM (m:n)" means that LLM was fine-tuned on a dataset where the ratio of theoretical data to real-user data is m/n.
>
> | Overall MSE        | Qwen2 | Llama-3.1 |
> | ------------------ | ----- | --------- |
> | Pre-SFT LLM        | 3.97  | 4.08      |
> | Mix-SFT LLM (1:10) | 2.85  | 3.17      |
> | Mix-SFT LLM (1:1)  | 2.38  | 1.76      |
> | Mix-SFT LLM (10:1) | 2.03  | 1.64      |
> | InvestAgent        | 2.16  | 1.59      |
>
> Table 2R: Comparison of the overall MSE between pre-SFT LLMs', Mix-SFT LLMs', and InvestAgents' investment decisions with real-user data in P1 (relative herd behavior). "Mix-SFT LLM (m:n)" means that LLM was fine-tuned on a dataset where the ratio of theoretical data to real-user data is m/n.
>
> | Overall MSE        | Qwen2 | Llama-3.1 |
> | ------------------ | ----- | --------- |
> | Pre-SFT LLM        | 17.22 | 13.07     |
> | Mix-SFT LLM (1:10) | 11.33 | 10.68     |
> | Mix-SFT LLM (1:1)  | 9.65  | 8.98      |
> | Mix-SFT LLM (10:1) | 7.32  | 7.06      |
> | InvestAgent        | 7.46  | 7.25      |
>
> From the above experimental results, it can be observed that adding a portion of real-user data slightly improved the model's performance on average, i.e., InvestAgents align more with real-user data, indicating that this approach can enhance the model's robustness to some extent. Notably, as the proportion of real-user data in the entire SFT training dataset gradually increases, the robustness may improve, but the parameter convergence rate decreases. We have provided both theoretical and experimental evidence for this in Section 4.2.
>
> ---
>
> **W2:** Thanks for the reviewer's comments! We conduct the experiment using the FinGPT datasets, including FinGPT-fineval [1] and FinGPT-convfinqa [2], to fine-tune LLMs, and compare them with our proposed InvestAgent. The experiment results are as follows.
>
> Table 3R: Comparison of the overall MSE between pre-SFT LLMs', FinEval-SFT LLMs', ConvFinQA-SFT LLMs' and InvestAgents' investment decisions with real-user data in P2 (absolute herd behavior).
>
> | Overall MSE       | Qwen2 | Llama-3.1 |
> | ----------------- | ----- | --------- |
> | Pre-SFT LLM       | 3.97  | 4.08      |
> | FinEval-SFT LLM   | 3.35  | 3.28      |
> | ConvFinQA-SFT LLM | 2.77  | 1.96      |
> | InvestAgent       | 2.16  | 1.59      |
>
> Table 4R: Comparison of the overall MSE between pre-SFT LLMs', FinEval-SFT LLMs', ConvFinQA-SFT LLMs' and InvestAgents' investment decisions with real-user data in P1 (relative herd behavior).
>
> | Overall MSE       | Qwen2 | Llama-3.1 |
> | ----------------- | ----- | --------- |
> | Pre-SFT LLM       | 17.22 | 13.07     |
> | FinEval-SFT LLM   | 13.74 | 11.16     |
> | ConvFinQA-SFT LLM | 10.86 | 9.61      |
> | InvestAgent       | 7.46  | 7.25      |
>
> From the experimental results above, it can be seen that InvestAgent outperforms the LLMs fine-tuned on the FinGPT datasets. This is because InvestAgent's training dataset is specifically constructed for studying optimal investment problems with herding behavior, whereas FinGPT is more general. Therefore, InvestAgent shows better performance in the context of optimal investment analysis.
> Moreover, we are currently exploring RLHF to further enhance the alignment of our InvestAgent. However, as a preliminary work, we have only conducted initial experiments due to time constraints. In future work, we plan to investigate the impact of RLHF on the InvestAgent model and the InvestAlign method more thoroughly. We will also compare it systematically with SFT to evaluate the effectiveness of different alignment strategies in investment decision-making tasks.
>
> [1] https://huggingface.co/datasets/FinGPT/fingpt-fineval.
>
> [2] https://huggingface.co/datasets/FinGPT/fingpt-convfinqa.

---

> > ### Author Response · Authors · 2024-11-19
> > **Response to Weaknesses 3 & 4**
> >
> > **W3:** Thanks for the reviewer's comments, and we appreciate the reviewer's insightful comment. We agree that focusing exclusively on herd behavior may limit the generalizability of our findings to other types of investor biases, such as overconfidence or recency bias. In behavioral finance, various cognitive biases (e.g., overconfidence, recency bias) and decision-making biases (e.g., risk aversion, herd behavior) are well-documented, explaining investor behavior beyond traditional economic models. However, a unified framework encompassing all these theories is still lacking, making it challenging to model them simultaneously. In this study, we use herd behavior as an initial example to explore the impact of decision-making biases on investor strategies. Specifically, we examine how herd behavior, reflected by metrics like the average deviation $D(P, Q)$ and herd coefficient $\theta$, influences optimal investment decisions. Moving forward, we plan to extend our analysis to other biases, such as overconfidence and recency bias, to make our model more comprehensive. Moreover, our contribution goes beyond examining specific behavioral finance theories; we propose a methodology to train LLMs using synthetic and real-user data to align models with various investor behaviors. This approach is not limited to herd behavior but can be adapted to study different biases, potentially providing a flexible framework for broader applications in behavioral finance.
> >
> > ---
> >
> > **W4:** Thanks for the reviewer's comments! We agree with the reviewer that the simplified model may not capture the complexity of larger institutional investor decisions or differences in risk tolerance. In this study, according to the assumption in the prior work in [1], we assume a simplified investment scenario focusing on the unidirectional influence from an investment assistant to an investor, specifically analyzing the impact of herding behavior. For instance, in an online social network, a leading expert with significant investment experience can serve as the investment assistant, while the followers are the investors. **When extending this concept to institutional and retail investors, the institutional investor may act as the investment assistant, and the retail investor corresponds to the follower in our model.** In the current setup, the decisions of the investment assistant are considered exogenous, modeled using the theoretical solution from the Merton problem [2], a classical approach in finance. We have accounted for the assistant's (the institutional investor's) risk aversion by incorporating a risk aversion coefficient $\alpha$. However, this model does not capture more complex behaviors such as overconfidence or mutual herding effects among institutional investors. In future work, we plan to enhance the generalizability of our approach by exploring scenarios with mutual influence among multiple investors. This will include modeling the decision-making processes of both institutional and retail investors using various behavioral finance theories, beyond the unidirectional influence assumption. We aim to better capture the complexity of decision-making in different segments of the financial market across different types of investors.
> >
> > [1] Wang, Huisheng, and H. Vicky Zhao. "Optimal Investment with Herd Behaviour Using Rational Decision Decomposition." 2024 43rd Chinese Control Conference (CCC). IEEE, 2024.
> >
> > [2] Merton, Robert C. "Lifetime portfolio selection under uncertainty: The continuous-time case." The Review of Economics and Statistics (1969): 247-257.

---

> > > ### Author Response · Authors · 2024-11-19
> > > **Response to Questions 1 & 2**
> > >
> > > **Q1:** Thanks for the reviewer's comments, and we appreciate this insightful question. Indeed, in behavioral finance, various cognitive biases such as overconfidence and recency bias significantly impact investor decision-making. We believe that the InvestAlign approach could be extended to incorporate these biases. However, this would require several adaptations, including changes to the modeling framework, analytical approach, and experimental design. Here, we take **overconfidence** as an example.
> > >
> > > 1) **Model Replacement:** To model overconfidence, we would need to replace this with a framework that captures investors' overestimation of their own abilities or the precision of their private information. A common approach is to adjust the expected returns in the utility function, where the investor perceives an inflated mean return due to overconfidence. If an investor exhibits overconfidence, the expected return $\mu'$ they use in decision-making could be modeled as $\mu'=\mu+\delta$, where $\delta>0$ represents the overestimation error. This adaptation requires modifying the drift term in the geometric Brownian motion to incorporate this biased perception, altering the investment strategy accordingly.
> > >
> > > 2) **Analytical Approach:** The analytical solutions for optimal strategies would also need to be adapted. For instance, under overconfidence, the perceived risk is often underestimated, leading to higher leverage in investment decisions. Therefore, we would need to derive new optimal decision rules considering this altered perception. In practical terms, this could involve modifying the optimal portfolio weights derived from the Merton problem to account for biased expectations.
> > >
> > > 3) **Experimental Data:** Extending InvestAlign to overconfidence would also necessitate changes in the dataset used for fine-tuning and evaluation. Specifically, we would require data reflecting scenarios where overconfidence is prevalent. This could be achieved by incorporating historical market data during periods characterized by high investor overconfidence or by designing user experiments where participants' confidence levels are intentionally varied and recorded. We could use experimental data where participants make predictions about asset returns. By comparing these predictions with actual market performance, we can quantify the level of overconfidence and use this as an input feature when fine-tuning the LLM.
> > > In future work, we plan to extend the InvestAlign framework by incorporating these cognitive biases systematically. We will develop specific adjustments for the modeling of overconfidence and other biases, derive the corresponding analytical solutions, and conduct empirical experiments to validate these extensions. By doing so, we aim to build a more comprehensive alignment methodology that accounts for a broader spectrum of investor behaviors in line with behavioral finance theories.
> > >
> > > ---
> > >
> > > **Q2:** Thanks for the reviewer's comments! We have explained and analyzed the experimental results in **W1** and **W2**.

---

> > > > ### Author Response · Authors · 2024-11-19
> > > > **Response to Question 3**
> > > >
> > > > **Q3:** Thanks for the reviewer's comments! The current investment scenario in our study models an investor making optimal decisions under the unidirectional influence of an investment assistant, with a focus on herding behavior. In an online social network context, a leading expert serves as the investment assistant, while the followers are the individual investors. Extending this model to different investor types, such as retail and institutional investors, involves specific adaptations.
> > > >
> > > > 1) **Model Assumptions and Structure:** Institutional investors often have more sophisticated decision-making processes, access to better information, and higher risk tolerance compared to retail investors. In our current model, we treat the investment assistant's decisions as **exogenous**, based on the Merton problem [1]. However, for institutional investors, this assumption might not hold. Institutional investors typically perform their own rigorous analysis and may influence market prices through their large trades. Therefore, adapting the model for institutional investors would require. Instead of treating the investment assistant's (institutional investor's) strategy as exogenous, we would need to model their decision-making process based on their market predictions, internal analyses, and behavioral biases, such as overconfidence.
> > > >
> > > > 2) **Incorporating Complex Decision-Making Biases:** Institutional investors may exhibit different biases compared to retail investors. For instance, overconfidence is more prevalent in institutional investors due to their perceived expertise and access to superior information. To adapt InvestAlign for institutional investors, we would need to modify the utility function or the decision-making framework to reflect this overconfidence. Adjust the expected returns used by institutional investors, incorporating an overconfidence parameter $\delta$, so that their perceived return $\mu'=\mu+\delta$. This adjustment would reflect their tendency to overestimate returns, impacting their portfolio choices and risk exposure.
> > > >
> > > > 3) **Market Segment Characteristics:** Institutional investors operate in different segments of the financial market (e.g., equity, fixed income, derivatives), each with distinct characteristics. Institutional investors may face liquidity constraints and market impact due to the size of their trades. This can be integrated into the model by adjusting the cost functions or incorporating a penalty term that captures the impact of large transactions on market prices.
> > > >
> > > > 4) **Experimental Data and Validation:** Extending the applicability of InvestAlign to institutional investors would also require specialized datasets reflecting their trading behaviors. Institutional trades often differ in size, timing, and strategy compared to retail trades. We would need to collect or simulate data specific to institutional strategies, possibly using transaction data from hedge funds, mutual funds, or pension funds. Historical data from institutional investor trades, such as 13F filings in the U.S., could be used to analyze their decision patterns, and this data could help fine-tune and validate the model for institutional settings.
> > > >
> > > > In future work, we plan to extend our framework by modeling mutual influences among multiple investors and incorporating a broader set of behavioral finance theories. This will help enhance the generalizability of our method, not only across different segments of the financial market but also to other high-stakes decision-making domains beyond finance.
> > > >
> > > > [1] Merton, Robert C. "Lifetime portfolio selection under uncertainty: The continuous-time case." The Review of Economics and Statistics (1969): 247-257.

---

> > > > > ### Author Response · Authors · 2024-11-19
> > > > > **Response to Question 4**
> > > > >
> > > > > **Q4:** Thanks for the reviewer's comments! A clear derivation of the theoretical model is shown in the prior work in [1], and we can present the specific assumptions here.
> > > > >
> > > > > 1) **Price Information:** The parameters of asset prices are known.
> > > > > - Risk-free Asset: $dB(t) = r B(t) dt$
> > > > > - Risky Asset: $dS(t) = \mu S(t) dt + \sigma S(t) dW(t)$
> > > > > 2) **Investment Purpose:** The investor only engages in investment activities.
> > > > > - Wealth Equation: $dX(t) = [r X(t) + v P(t)] dt + \sigma P(t) dW(t)$
> > > > > 3) **Utility Function:** The investor has CARA (Constant Absolute Risk Aversion) utility functions.
> > > > > - Utility Function: $\phi [X(T)] = -\alpha^{-1} e^{-\alpha X(T)}$
> > > > > 4) **Herd Behavior:** There is herd behavior between the user and the assistant.
> > > > > - Absolute Decision Distance: $D(P, Q) = \frac{1}{2} \int_0^T [P(t) - Q(t)]^2 dt$
> > > > > - Relative Decision Distance: $D(P, Q) = \frac{1}{2} \int_0^T [P'(t) - Q'(t)]^2 dt$
> > > > > 5) **Network Structure:** The assistant unidirectionally influences the investor's decision-making.
> > > > > - User's Objective: $J(P) = \mathbb{E}[\phi (X(T))] - \theta D(P, Q)$
> > > > > 6) **Assistant's Decision:** The assistant's decision is exogenous [2].
> > > > > - Assistant's Decision: $Q(t) = (\mu - r) / (A * \sigma^2) * \exp(r * (t - T))$
> > > > >
> > > > > The assumptions in the above model, while useful for theoretical analysis, may limit its alignment with real investor behavior as indicated by various studies in the economic literature. For instance, the assumption of known asset price parameters simplifies the investment problem but overlooks the reality of imperfect information where investors frequently update their beliefs based on new data [3]. Moreover, focusing solely on investment without considering other financial needs (e.g., consumption or liquidity requirements) fails to capture the broader objectives that influence real investor decisions [4]. The use of CARA utility functions, which implies constant risk aversion, may not reflect the observed variability in risk preferences, as empirical evidence shows that investors often exhibit decreasing absolute risk aversion [5]. Additionally, our model's assumption of absolute herd behavior between the user and the leader overstates the degree of herding behavior in practice, where investor decisions are typically influenced by a mix of independent judgment and partial herding [6]. Finally, the unidirectional influence from leader to user simplifies the dynamic interactions observed in financial markets, where influence is often bidirectional [7]. Given these limitations, the model in [1] may not fully capture the complexity of real-world investor behaviors and decision-making processes. However, to validate whether the assumptions of the theoretical model fit the real world well is out of scope in our ICLR work. We plan to investigate extensions of the model by relaxing these assumptions, incorporating factors like dynamic risk aversion, imperfect information, and mutual influence among investors to better reflect real market conditions and align with behavioral finance theories in future work.
> > > > >
> > > > > [1] Wang, Huisheng, and H. Vicky Zhao. "Optimal Investment with Herd Behaviour Using Rational Decision Decomposition." 2024 43rd Chinese Control Conference (CCC). IEEE, 2024.
> > > > >
> > > > > [2] Merton, Robert C. "Lifetime portfolio selection under uncertainty: The continuous-time case." The review of Economics and Statistics (1969): 247-257.
> > > > >
> > > > > [3] Grossman, Sanford J., and Joseph E. Stiglitz. "On the impossibility of informationally efficient markets." The American Economic Review 70.3 (1980): 393-408.
> > > > >
> > > > > [4] Campbell, John Y., and Luis M. Viceira. Strategic Asset Allocation: Portfolio Choice for Long-Term Investors. Oxford University Press, 2002.
> > > > >
> > > > > [5] Fama, Eugene F., and Kenneth R. French. "Common risk factors in the returns on stocks and bonds." Journal of Financial Economics 33.1 (1993): 3-56.
> > > > >
> > > > > [6] Bikhchandani, Sushil, and Sunil Sharma. "Herd behavior in financial markets: A review." IMF Staff Papers 47.3 (2000): 279-310.
> > > > >
> > > > > [7] Shiller, Robert J. "Stock prices and social dynamics." Brookings Papers on Economic Activity 1984.2 (1984): 457-510.

---

> > > > > > ### Comment · Reviewer_uBAV · 2024-11-27
> > > > > >
> > > > > > Thank you for your thorough response. I see that extending the model to capturing other behavior or generalize it could be doable, but it requires new work. The indicates that the framework is still somewhat limited in the scope and generalizability. Thus, I will retain my current rating.

---

### Author Response · Authors · 2024-11-19
**General Response**

Dear Reviewers,

We sincerely appreciate the insightful critiques and valuable feedback from the reviewers. Addressing the key concerns raised, we would like to highlight the most critical issues and our corresponding revisions as follows.

1. **Generalizability and Robustness:** This has been identified as a paramount concern across all reviews (WAhM, MvGL, 7wmn, uBAV). To bolster the robustness and applicability of our theoretical model, we have taken significant steps. Firstly, we have supplemented our theoretical data with smaller samples of real-user data to enhance the model's robustness, as suggested in uBAV's review (AW1). This hybrid approach not only maintains the advantages of theoretical data in terms of parameter convergence but also grounds the model in real-world investor behavior. Additionally, we have expanded our comparison to include other alignment strategies, such as training on existing behavioral finance datasets, which were recommended by uBAV (AW2). This comparison underscores the orthogonal nature of our method to traditional Supervised Fine-Tuning (SFT), and how it can complement these methods.

2. **Clarification of Study Scope and Objective:** A key point we want to emphasize is the scope of our study. Our goal is not to propose a universal investment language model or a fully capable agent that can solve every complex financial decision-making problem. Instead, our primary objective is to address a specific challenge: data scarcity when training large language models (LLMs) in the context of investor decision-making. Real-world investor data is often limited due to privacy concerns and high collection costs, making it difficult to build extensive datasets for training. Thus, our work focuses on exploring a new potential method to overcome this data scarcity issue by constructing training datasets based on simplified problems with theoretical solutions. By using this approach, we aim to provide a feasible pathway for aligning LLMs with human-like decision-making behaviors in scenarios where real data is insufficient. This differentiation is crucial, as it highlights that our paper targets solving the data scarcity problem rather than developing a universal agent for all investment decision-making tasks.

3. **Assumptions of the Theoretical Model:** uBAV and WAhM questioned the assumptions underlying our theoretical model. We have provided a detailed explanation of these assumptions and their implications on model alignment with real investor behavior in uBAV's review (AQ4) and WAhM's review (AW1). We acknowledge the limitations and plan to relax these assumptions in future work to better reflect real market conditions.

4. **Applicability to Different Types of Investors and Behavior Biases:** uBAV inquired about the applicability of InvestAlign to various investor types and behavior biases. We have clarified that our current model focuses on a simplified investment scenario and plan to extend our analysis to other biases and more complex decision-making processes in future work, making our model more comprehensive (AW3 and AW4).

5. **Insufficient Real User Data:** MvGL pointed out the limitation of our real user data sample size. In response, we have increased our sample size to ensure that our findings are more representative and robust, aligning with statistical testing recommendations (AW2).
In summary, we have made substantial efforts to address the reviewers' concerns, particularly focusing on enhancing the generalizability and robustness of our theoretical model, which was the most critical feedback. We believe these revisions significantly strengthen our paper and its contributions to the field of aligning LLMs with investor decision-making processes.

Sincerely,

The Authors

---

### Meta-Review · Area_Chair_2M6E · 2024-12-20

**Metareview:**

This paper aims to align large language models (LLMs) with real investor decision-making processes under herd behavior in financial markets. Reviewers agreed that this paper studies a very interesting and challenging problem and makes contributes to the community by collecting an extensive real-world data, presenting a multi-round experimental design, etc. However, reviewers also identified some limitations of this work, such as the lack of comparisons with other LLM alignment techniques, the design of tasks P1 and P2, generalization ability of the proposed approach, etc. Overall, this paper in its current version is not ready for publication at ICLR.

**Additional Comments On Reviewer Discussion:**

Reviewers raised concerns on task designs, baselines, technical details, etc. The authors provided responses with additional results, which  have partially addressed the concerns from reviewers. However, some major concerns still remain, such as the design and justifications of P1 and P2 in practice.

---

### Decision · Program_Chairs · 2025-01-22

Reject